# Room-temperature second sound in isotopically pure graphite

Zhikun Xie [1,2,3,4,9], Yifan Zhang[1,2,3,4,9], Xin Huang [5,9], Zhiwei Ding [6], Jie Wei[1,2,3,4], Difei Dong[1,2,3,4], Kun Cao[1,2,3,4], Tianshu Lai [1,2,3,4], Kenji Watanabe [7], Takashi Taniguchi [7], Xin Qian[8], Masahiro Nomura [5] ✉ & Ke Chen [1,2,3,4] ✉

The observation of second sound—a propagating wave-like manifestation of hydrodynamic heat transport—in solid crystals has been confined to a handful of materials at cryogenic temperatures, as disorder and Umklapp scattering suppress this phenomenon at room temperature. Here, we report the direct observation of second sound at ambient conditions in isotopically purified graphite. Using transient thermal grating spectroscopy, we measure a distinct damped oscillatory signal that provides unambiguous evidence of second sound, decisively distinguishing it from diffusive and ballistic transport regimes. This collective phonon dynamics enables an enhancement of the effective thermal conductivity, even surpassing the conventional diffusive limit by nearly 10%. Our work establishes the control of phonon-isotope scattering as a powerful strategy to unlock hydrodynamic phonon transport. It demonstrates that phonon hydrodynamics is an accessible and exploitable phenomenon in crystals at room temperature, providing an avenue for the fundamental study and application of wave-like heat transport.

Second sound, a phenomenon characterized by the propagation of temperature waves in materials, is a hallmark of hydrodynamic heat transport. First discovered by V. Peshkov in 1944 in superfluid helium-3 (He-3) near 1.4–1.6 K[1], second sound provided pivotal experimental evidence for Landau's two-fluid model of superfluidity[2]. Unlike the conventional diffusive process described by Fourier's law of heat conduction, second sound was attributed to the coherent collective wave motion of thermal energy. The theoretical framework for second sound in solids was established through the work of R. Peierls and L. Landau, who suggested that under specific conditions—such as low temperatures and minimal resistive scattering—phonons could exhibit collective behavior akin to fluid dynamics. Indeed, early observations

of second sound in solids were all achieved near liquid helium temperatures, notably in bismuth (Bi) at 1.2–4.0 K[3] and sodium fluoride (NaF) at 10–18 K[4].

Significant progress has been made in the study of second sound recently, driven by advancements in experimental techniques and theoretical models. With the development of ultrafast laser-based methods and improved detection technologies, the temperature and material constraints of phonon hydrodynamic transport have been increasingly delineated. For examples, using transient thermal grating (TTG) technique, researchers have successfully measured second sound propagation in materials like graphite at temperatures above 100 K[5] and above 200 K[6], consecutively. With high frequency

[1]State Key Laboratory of Optoelectronic Materials and Technologies, School of Physics, Sun Yat-sen University, Guangzhou, China. [2]Guangdong Provincial Key Laboratory of Magnetoelectric Physics and Devices, School of Physics, Sun Yat-sen University, Guangzhou, China. [3]Center for Neutron Science and Technology, School of Physics, Sun Yat-sen University, Guangzhou, China. [4]Center for Advanced and Ultrafast Photonic Science, School of Physics, Sun Yat-sen University, Guangzhou, China. [5]Institute of Industrial Science, The University of Tokyo, Tokyo, Japan. [6]Department of Material Science and Engineering, Massachusetts Institute of Technology, Cambridge, MA, USA. [7]Research Center for Materials Nanoarchitectonics, National Institute for Materials Science, Tsukuba, Japan. [8]School of Energy and Power Engineering, Huazhong University of Science and Technology, Wuhan, China. [9]These authors contributed equally: Zhikun Xie, Yifan Zhang, Xin Huang. ✉e-mail: nomura@iis.u-tokyo.ac.jp; chenk35@mail.sysu.edu.cn

modulation ( ~ 100 MHz) of the heating source, driftless second sound was inferred in Germanium by its frequency domain thermore-flectance (FDTR) signal[7]. The phonon Peierls-Boltzmann transport equation (PBTE) and equilibrium correlation functions have clarified the theoretical description of second sound as damped temperature waves[8,9], and a bunch of materials have been predicted to support phonon hydrodynamic transport at relatively high temperatures[10,11]. Recent explorations of phonon hydrodynamics continue to challenge conventional paradigms of heat transport. For examples, by making the thickness of graphite bar thin to 8.5 micrometers, a record high room-temperature thermal conductivity of 4300 W/m·K, attributed to phonon hydrodynamics, was measured[12]. And by design and fabrication of special structures from isotope-enriched graphite, novel phonon Poiseuille flow in micron-wide ribbons[13], and the rectification of phonon heat conduction via Tesla valve[14], have been successfully demonstrated, opening new avenues for energy-efficient device design and advanced thermal regulation techniques by phonon hydrodynamic behaviors.

There is growing interest in pushing second sound observations toward higher temperatures in thermal transport research[7,10,12,15]. Effective scheme to stimulate phonon hydrodynamic transport can root back to the physics of phonon scattering. While normal phonon scattering, where the wavevectors of the involved phonons lie in the first Brillouin zone (the Wigner–Seitz cell of the reciprocal lattice), maintains the overall flow of phonon crystal momentum; resistive scattering, such as Umklapp phonon scattering, phonon-electron scattering, phonon-defect scattering, can disrupts the total phonon momentum and reduce the overall heat conduction. Specifically, for second sound to manifest, normal scattering needs to be substantially stronger than resistive scattering. Isotope atoms in a crystal can be treated as defect or impurity, thus phonon-isotope scattering is one source of the resistive scatterings. Purification of isotopic concentration diminishes phonon-isotope scattering, which could dramatically enhance thermal conduction and facilitate the occurrence of phonon hydrodynamics. For examples, 90% of thermal conductivity increase has been demonstrated in isotope enriched cubic boron nitride compared to the natural isotope counterpart[16]. And isotope-enrichment in graphite plays a crucial role in achieving the phonon Poiseuille flow and the phonon Tesla valve[13,14] mentioned above., this research aims to explore the potential of isotope enrichment of graphite material to fulfill the emergence of second sound at room temperature. Specifically, we synthesize graphite materials with ultrapure isotope concentration, inspect the existence of collective wave behavior of thermal excitation in the purified material with the TTG method, and perform the same experiments on graphite piece with natural isotope composition for comparison. The smoking gun of second sound is discovered in the isotope-purified graphite, which demonstrates that second sound can appear at room temperature as a result of the giant isotope effect. An important impact of phonon hydrodynamics on thermal conductivity is investigated and revealed as well.

## Results and discussion

The isotope-enriched graphite samples were grown using high pressure and high temperature method[17]. Millimeter scale small thin flakes with thickness of ~150 μm, were synthesized. The resulting synthesized graphite possesses a high isotopic purity of 99.93% $^{12}C$, providing an ideal platform to minimize extrinsic resistive scattering. For comparison, we commercially obtained a high-quality highly oriented pyrolytic graphite (HOPG) with natural isotopic abundance as the reference sample. Microscope images reveal a typical flat and uniform surface area (Fig. 1a), large enough for our optical transport measurements. Secondary ion mass spectroscopy (SIMS) measurement yields that the average 12 Carbon isotope concentration of the purified samples reaches as high as 99.93%, with only 0.07% $^{13}C$ remaining

(inset of Fig. 1b). The result of Raman spectroscopy reflects the isotope composition difference between the purified sample (0.07% $^{13}C$) and the unpurified HOPG (1.10% % $^{13}C$), by a reasonable blue shift of Raman peak in the purified graphite since its slightly lighter atomic mass (Fig. 1b). The well-established transient thermal grating (TTG) technique was utilized to study the dynamics of the excited thermal energy. In TTG, two pump laser pulses overlap and interfere at the sample surface, creating an initial sinusoidal thermal energy (temperature) grating distribution (Fig. 1c, upper). This thermal grating induces thermal expansion in graphite. When a third probe laser pulse hits the graphite surface, it is diffracted by the sinusoidal thermal expansion grating. A fourth called "reference" laser pulse also hit the sample surface, arriving at the same time as the probe one. Due to the ingenious design of the TTG optical system, the reflection of the reference beam would automatically coincide with the diffracted probe beam in space (Fig. 1c, lower, 0 Ref. & -1 Probe), enabling the amplified heterodyned detection. Such spatially overlapped probe diffraction and reference reflection is taken as the signal, whose temporal intensity variation reflects the evolution of the transient grating.

TTG essentially probes the difference in quasi-temperature (i.e., the collective phonon energy) between the interference peak and valley positions[5,6], while remaining generally unaffected by the thermalization of the acoustic phonons (see Methods and Supplementary Fig. 1 for details). This brings one major advantage of TTG in the study of phonon transport: it presents distinct temporal signal shapes for different phonon transport regimes, such that one can easily identify what the phonon motion is under the experimental condition by simply investigating the signal pattern. For example, diffusive phonons flatten the thermal grating and produce a TTG signal that decays exponentially; But ballistic phonons, with different wavevectors and velocities, result in a TTG signal exhibiting non-periodic oscillation with a positive background[6,18]; However, hydrodynamic phonons make the thermal grating evolve as a damped standing wave, and yield a damped oscillatory TTG signal[6,8] (see Supplementary Movie 1); Therefore, a damped oscillation in the TTG measurement, is recognized as the hallmark of second sound phenomenon. Our TTG setup provides ~100 fs temporal resolution, which is sufficient to capture the ultrafast dynamics, and a tunable grating period (range from 0.95 μm to 54 μm), covering the transition from ballistic to hydrodynamic and diffusive regimes.

We measure TTG signals for both the isotope-purified graphite and the HOPG, with a grating period of 0.95 μm, at room temperature (Fig. 1d). The signals are normalized for better comparison. While the signal of HOPG shows an exponential decay, indicating diffusive phonon transport in HOPG with natural isotopic abundance, the signal of the isotope-enriched graphite shows a quasi-damped oscillatory behavior, with the signal flipping from initially positive to later negative and gradually recovering as time evolves. According to above discussion, this sign-flipped signal means that the second sound thermal wave is occurring and propagating with damping in the isotopically pure graphite at room temperature (Supplementary Movie 1). The damping which comes from resistive scattering such as phonon Umklapp scattering, is expected to be relatively strong at room temperature, so that the oscillatory signal only appears with one negative dip but without the consecutive second positive peak due to the strong attenuation. Nevertheless, the dip depth remains a good indicator of the hydrodynamic nature of the thermal wave. By utilizing the state-of-the-art calculation techniques, i.e., combing the ab initio first principle calculation of graphite's phonon properties with a full scattering matrix formulism of PBTE solution[6,19], we simulate the TTG signals for both isotopically enriched and natural graphite (Fig. 1d), using $^{12}C$ concentration values of 99.93% and 98.90%, respectively, and without any fitting parameters. The full scattering matrix PBTE has been established as a validated and widely accepted theoretical framework for simulating phonon hydrodynamic transport[5,6,19,20], and a detailed

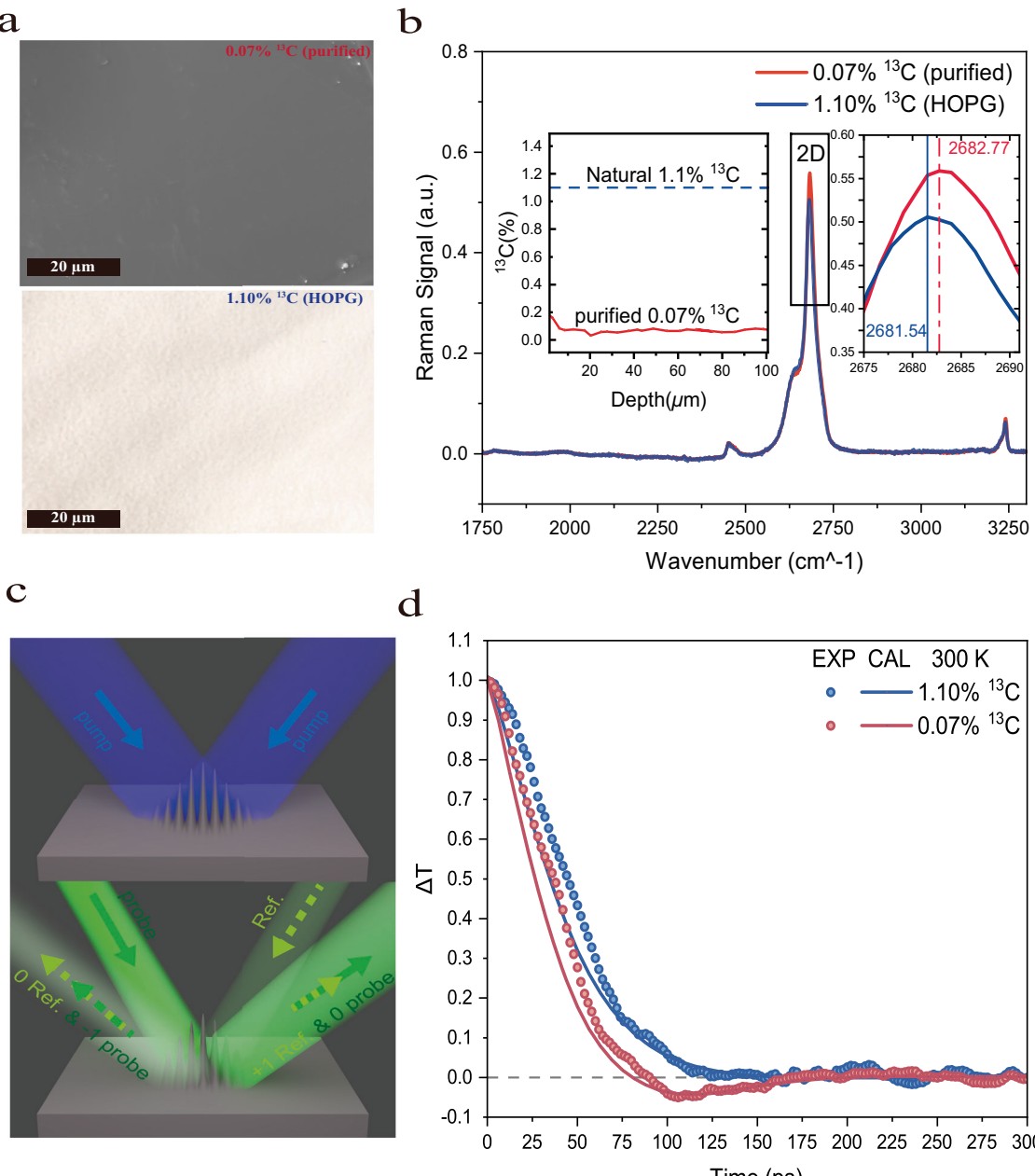

**Fig. 1 | Observation of second sound at room temperature in isotope-enriched graphite. a** The scanning electron microscopy image of isotope-enriched graphite samples (top) and optical microscope image of highly oriented pyrolytic graphite (HOPG)(bottom). **b** The Raman signal shows the isotopic abundance difference between the purified sample (0.07% $^{13}$C, red) and the unpurified HOPG (1.10% $^{13}$C, blue). Inset: SIMS measurement shows carbon concentration of isotope-enriched graphite samples. **c** Schematic illustration of the excitation and the detection of transient thermal grating. **d** The results of TTG signals at room temperature (300 K), with grating period of 0.95 μm.

description of this methodology is provided in the section of Details of Numerical Methods. Excellent agreement between the experiments and the simulation is achieved, with the reappearance of the hydrodynamic sign flipping feature for the isotope-enriched graphite and the diffusive decay feature for HOPG accordingly. Especially, the time position and the depth of the dip coincide well between the measured signal and the simulated one. This suggests that both the speed and the strength of second sound, can be mutually confirmed and acquired by the experiment and the PBTE theory. In a word, room temperature drifting second sound is observed in isotope-enriched graphite with TTG measurement, and the results can be well supported by first-principles calculation, demonstrating that the technologically-feasible isotope engineering in graphite (and potentially other materials)

provides a platform to explore the hydrodynamic wave-like heat transport at room-temperature.

For second sound to appear, there is one requirement: the characteristic transport length L (half of the grating period in the TTG case) must be larger than the MFP due to normal scattering but smaller than the MFP due to resistive scattering, i.e., $MPF_N < L < MPF_R$. The temperature and the grating period can affect the satisfaction of this requirement, and thus the emergence of second sound, giving us a good way to reveal the transition between phonon hydrodynamic behavior and phonon diffusive motion around room temperature with different transport lengths (grating periods) (Fig. 2). Each sub-figure of Fig. 2 displays the experimental and the simulated TTG signals under three temperatures. The experiments and the simulations agree well, showing

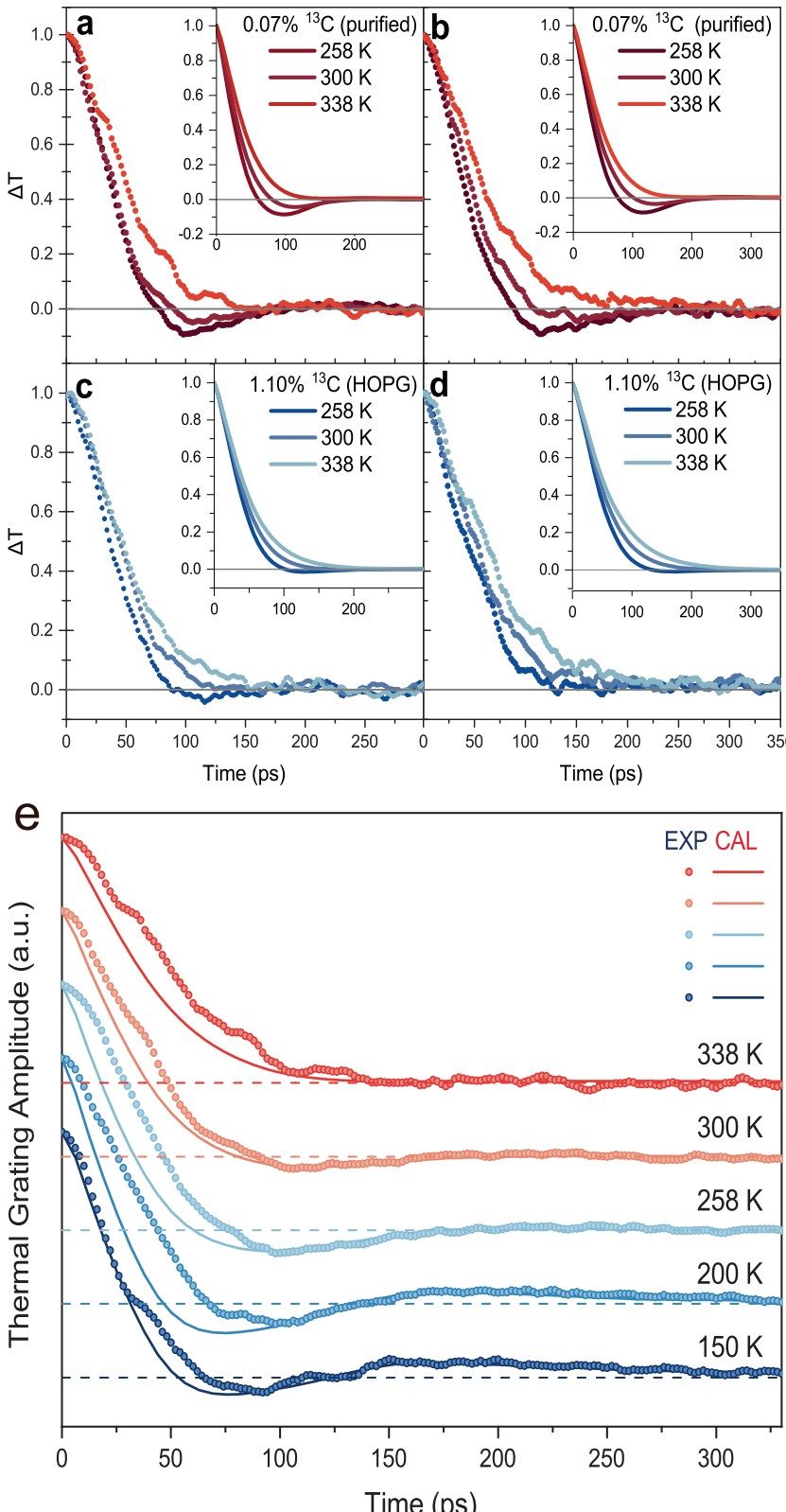

**Fig. 2 | Experimental and simulated TTG dynamics. a–d** Measured (dots) and simulated (curves) TTG signals for isotope purified graphite (0.07% $^{13}$C, a and b) and HOPG (1.10% $^{13}$C, **c**, **d**). Horizontal gray lines indicate zero for each pair of curves. The left and right columns respectively show the TTG signals for grating periods grating period of 0.95 μm (a and c) and 1.1 μm (**b**, **d**). **e** Measured (dots) and simulated (curves) TTG signals for isotope purified graphite with periods grating period of 0.95 μm at different temperatures.

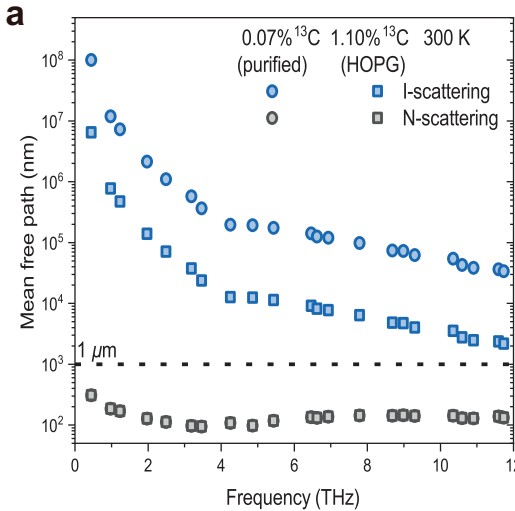

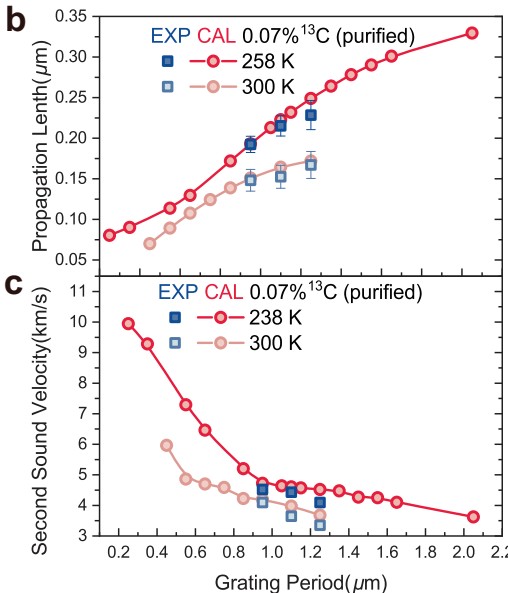

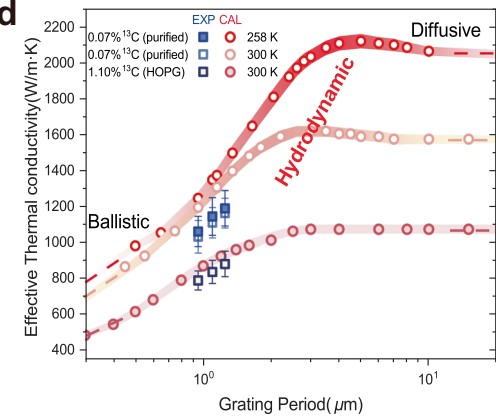

**Fig. 3 | Effective thermal conductivity enhancement by hydrodynamic phonons. a** Comparison of mean free path for isotope-enriched graphite samples (squares) and HOPG (dots) at 300 K, I-scattering and N-scattering. **b, c** The propagation length and the second sound velocity extracted from the measured (blue dots) and the simulated (red curves) TTG signals. **d** At 258 K and 300 K, the effective thermal conductivity from measured (blue squares) and simulated (red dots) depend on the grating period. The red bands show the trend of the simulated effective thermal conductivity, which divides into the diffusion region, hydrodynamic region, and ballistic region as the grating period decreases. The areas where the red bands are concentrated highlight the enhancement of the effective thermal conductivity due to hydrodynamic phonons.

are compared, as shown in Fig. 2e. When temperature is low enough, not only the dip becomes deeper, but also the signal can rebound to the positive region. The temperature effect on the emergence of second sound is expected, as the temperature decreases, phonons tend to concentrate on small-wavevector modes, therefore the resistive scattering of phonon Umklapp scattering which in principle involves large-wavevector phonons becomes reduced. Setting aside possible ballistic phonon effect, the lower the temperature, the better the condition $MPF_N < MPF_R$ holds, and hence the characteristic of second sound becomes more pronounced (Fig. 2e and Supplementary Fig. 2). Isotope purification can further reduce the resistive scattering. Specifically, the presence of $^{13}C$ isotopes acts as point defects that induce resistive phonon-isotope scattering. In natural graphite (1.10% $^{13}C$), this scattering is strong enough to relax phonon momentum, thereby preventing the formation of second sound at room temperature. By reducing the $^{13}C$ concentration to 0.07%, this scattering channel is effectively suppressed. The calculated phonon-isotope scattering rate in HOPG is found to be about one order-of-magnitude larger than that of isotope-enriched graphite (Fig. 3a). This significant difference explains why phonon hydrodynamic thermal waves are much less likely to be observed in HOPG at room temperature.

As the grating period increases, for HOPG, the decay rate of the TTG signals get slower, due to the fact that in the diffusive transport the decay rate is proportional to $Dq^2$, where $D$ is the thermal diffusivity and $q$ (= $2\pi/L_g$) the grating wavevector; While for the isotope-enriched graphite, the dip position increases and the dip depth decreases. The dip position denotes the time that the second sound wave has just traveled a distance of half of the grating period. Thus, larger grating period means larger distance to for second sound to propagate, leading to a larger dip time position. Meanwhile, larger grating period also leads to poorer satisfaction of the hydrodynamic requirement $MPF_N < L_g < MPF_R$. As transport length $L_g$ grows to comparable and even larger than $MPF_R$, the thermal wave nature–the dip depth becomes weaker and even disappears, with TTG signal trending to an exponential decay in the diffusive regime, as shown by the case of 258 K 1.25 μm HOPG in Fig. 2d.

Quantifying the effective thermal conductivity ($k$) in the spatial-temporal scale relevant to transient phonon hydrodynamic regime is an important task[18], but it has been still seldom investigated by the experimentalists. TTG, again, provides a convenient experimental method to investigates transport properties of second sound and determine the effective thermal conductivity as a function of transport length. As mentioned above, the dip position $t_d$ tells us the time the thermal wave takes to travel half of the grating period $L_g$, from which the second sound propagation speed $v_{ss}$ (= $0.5L_g/t_d$) is known, and the dip depth $\Delta T_d$ tells us how much attenuation the thermal wave decays at that dip time, from which the second sound propagation length, i.e., the traveling distance corresponding to the 1/e amplitude, $l_{ss}$ (= $-0.5L_g/\ln(-\Delta T_d)$) is known. As for heat dissipation, fitting the TTG signal with an exponential decay yields the attenuation rate $\Gamma$ of the peak-valley temperature difference $\Delta T$ (Supplementary Fig. 3), from which the effective thermal conductivity can be obtained by the formula[5,21,22]: $k = c\Gamma/q^2$, where $c$ is the volumetric heat capacity of the graphite

the same trending pattern during the temperature or grating period changes. As the temperature decreases, the TTG signals decay faster correspondingly. As shown in isotope-enriched graphite at 300 K and 258 K, the TTG signals decay and cross to the negative region, forming the dip of second sound signature. The lower the temperature, the deeper the dip, indicating stronger hydrodynamic feature. This trend becomes even more evident when additional low-temperature signals

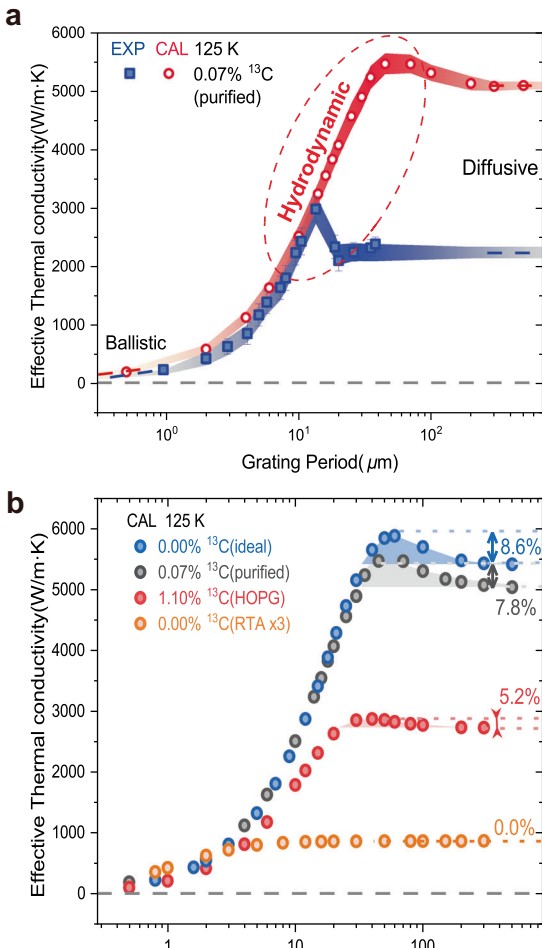

**Fig. 4 | Thermal conductivity enhancement by hydrodynamic phonons.**
**a** Thermal conductivity enhancement of isotope purified graphite by hydrodynamic phonons at 125 K. **b** Thermal conductivity enhancement as a function of isotope concentration. The blue, black, and red dots represent the effective thermal conductivities for three different isotope concentrations: ideal isotopically pure graphite (0.00% $^{13}$C), the purified sample (0.07% $^{13}$C), natural abundance HOPG (1.10% $^{13}$C), and pure graphite (0.00% $^{13}$C) calculated from the PBTE within the single-mode relaxation time approximation (RTA) respectively. The RTA data in orange is multiplied by 3 and plotted for visibility.

value of Fourier's law when there is a temperature gradient over a distance shorter than the phonon MFP, as in the cases of short-grating-periods TTG measurements. This downward trend indicates that the phonon transport transits from (bulk) diffusive to (short) ballistic transport. However, one could notice that there is a little bump exceeding the bulk level in the effective $k$ result around 5 μm grating period and 3 μm grating period for 258 K and 300 K, respectively. And it is noteworthy that such bump can only appear in the measurements of isotopic enriched graphite but not in the case of HOPG. This is a strong indicator that the bump, i.e., the relative enhancement of the effective $k$, comes from the phonon hydrodynamic transport.

In order to verify that the $k$ enhancement effect is due to the hydrodynamic phonons, we further measured and simulated the TTG signals at a relatively low temperature of 125 K, where the phonon hydrodynamic feature is more prominent[5]. Some of the signals with typical shape are shown in Supplementary Fig. 2a (and Supplementary Fig. 2b for 1.10% $^{13}$C HOPG as well). As expected, the TTG signals of isotope purified graphite at low temperature indeed show much stronger dip (down to −0.2) and noticeable flipped positive part following the dip. At moderate low temperatures, such as 80-200 K for graphite, phonons mainly occupy the small-wavevector states according to Bose−Einstein distribution, therefore the diffusive Umklapp phonon-phonon scattering which in principle requires large wavevector is suppressed, but the desired normal phonon-phonon scattering can still keep, promoting the relative importance and the stronger behavior of the hydrodynamic transport. As shown in the inset of Supplementary Fig. 2a, even the HOPG supports prominent second sound wave at 125 K, simply with lower dip (−0.09) and weaker oscillation than the purified case.

We extracted the effective $k$ of the pure graphite by fitting the measured and the simulated result with exponential decay (Fig. 4a and Supplementary Fig. 2c). The values of the experimental results and the simulation are close for small grating periods, but the experimental $k$ starts to deviate from the theoretical one for grating periods larger than 10 μm, which we attribute to the transport limit due to grain boundary scattering in the pure graphite sample (see Supplementary Fig. 4b for the grain size information). However, the simulation is performed for an ideal crystal, so it is free from grain boundary scattering. At 125 K, the simulation tells that the bump in the effective $k$ not only still appears, at a longer grating period of 50 μm, but also shows a significantly larger magnitude, than the cases of 258 K and 300 K. Further calculations reveal that ideally purified isotopic graphite (0.00% $^{13}$C) exhibit a more pronounced bump with stronger second sound (Fig. 4b). For comparison, we also performed simulations based on the solution of PBTE under the single relaxation time approximation (RTA). It has been confirmed that PBTE under single RTA cannot capture phonon hydrodynamics[18]. As seen in Fig. 4b, the $k$ result of RTA does not show any bump but only the traditional size effect pattern. To summarize, the re-emergence of the $k$ bump at low temperature and purer sample with much stronger intensity, together with the disappear of the $k$ bump in the RTA case, confirms our inference: The phonon hydrodynamic transport can indeed increase the actual heat flux via constructive energy and momentum interactions, and lead to an enhanced effective thermal conductivity.

Strikingly, the peak value of the effective $k$ in the hydrodynamic regime can be 8.6% larger than its diffusive bulk level. While traditional size effect only predicts a lower effective $k$ at small transport scales than the bulk diffusive limit, such absolute $k$ enhancement above the bulk plateau has never been discovered in previous experiments. In the literatures, extensive studies have been devoted to the thermal conductivity of graphite and graphene[20,23,24]; however, the impact of hydrodynamic phonons on thermal conductivity enhancement has rarely been revealed and discussed[12], because sparse materials can support remarkable second sound, and it's difficult to acquire accurate measurements of second sound and thus determine the transient

material. Therefore, we can extract the speed and the propagation length from the measured and the simulated signals (Fig. 3b, c), and acquire the effective thermal conductivity accordingly (Fig. 3d). $l_{ss}$ and $v_{ss}$ are functions of the grating period and the temperature, manifesting the dispersion relation of second sound, as a result of the interplay among hydrodynamic phonons, ballistic phonons and diffusive phonons[6]. $l_{ss}$ decreases with decreasing grating period, because the ballistic phonons become stronger at smaller transport scale, and they do not interact or couple their energy together, so the second sound thermal packet becomes not that easily confined and gets spread more quickly at smaller scale. $v_{ss}$ increases with decreasing grating period, because the group velocity of ballistic phonons is larger than that of hydrodynamic phonons, with the ballistic component gets stronger, the second sound thermal wave can travel faster.

As shown in Fig. 3d, the effective thermal conductivity decays from the bulk value at around 10 μm to only 50% or below as the grating period decreases to several hundred nanometers. This behavior is the well-known thermal conductivity size effect, originating from the fact that the actual heat flux is smaller than the predicted

thermal transport property. Here, we exploited the isotope purification effect to fully realize the hydrodynamic potential of the transmitted phonons in graphite, utilized the well-established TTG scheme, and directly demonstrated the meaningful enhanced thermal transport supported by the fluid-like phonons at the appropriate intermediate "hydrodynamic" spatial scales.

Why can phonon hydrodynamic motion increase the effective thermal conductivity? We get the answer from a derivation of the Callaway-PBTE[25]. Callaway simplified the complex phonon collision term in PBTE into two terms, one for the momentum-conserving normal scattering to capture phonon hydrodynamics, and the other for the momentum-destroying resistive scattering to manifest phonon diffusion. This approach provides a clear and intuitive physical picture. Starting from the Callaway-modified phonon Boltzmann transport equation, we first derive two coupled viscous heat equations, analogous to those obtained from the linearized BTE within the relaxon framework. These viscous heat equations can then be further reduced to a hyperbolic form (or namely thermal telegraph equation)[6,26,27], which is formally identical to the Cattaneo–Vernotte (CV) equation[28,29]. Detailed derivation can be found in Methods section. At this point, we realize that the thermal conductivity is actually composed of two parts: $k = k_K + k_H$, where $k_K$ is the kinetic part corresponding to the phonon diffusion, while $k_H$ is the hydrodynamic part corresponding to phonon hydrodynamics. Our TTG measurements probe the transient thermal transport. When the grating period is in the appropriate intermediate range (e.g., 20–200 μm for 125 K), the probed timespan (roughly 1-50 ns) matches or only modestly exceeds the resistive scattering time (0-0.2 ns, see Supplementary Fig. 5a). Consequently, fast normal scatterings remain within the detection window, sustaining a drifting population and permitting the Callaway-PBTE framework to capture the hydrodynamic contribution. The result is a clear $k$ enhancement (Fig. 4b). However, when the grating period gets large enough (e.g., >200 μm for 125 K), the probed timespan (> 50 ns) far outlives the resistive scattering time. In this case, hydrodynamic signatures appear only at the earliest moments and are subsequently erased by resistive damping, leaving the long-time average indistinguishable from diffusion. The Callaway-PBTE then collapses to the single-relaxation RTA-PBTE, $k_H$ vanishes, and the transient effective conductivity settles to the diffusive plateau (Fig. 4b).

Our work unveils the important role of isotopic purity in thermal transport, providing clear experimental evidence for the existence of second sound at room temperatures, and a notable "super-diffusive" enhancement in thermal conductivity driven by hydrodynamic phonons. The discovery here contributes to a deeper physical understanding of phonon dynamics, and provides opportunities for applications that leverage this unique fluid-like thermal transport mechanism in energy-efficient materials and devices.

## Methods

### Sample characterization

The $^{12}C$ and $^{13}C$ isotope concentration of our isotope-enriched graphite was measured using Time of flight-secondary ion mass spectroscopy (TOF-SIMS) to be 99.93% and 0.07%, respectively, as shown in Fig. 1b. Our natural isotope graphite is highly oriented pyrolytic graphite (HOPG) of the highest quality commercially available grade with relatively flat areas (HOPG Grade 1, SPI). According to the manufacturer, the mosaic spread of the sample is 0.4 ° ± 0.1 °. As shown in Supplementary Fig. 4a, electron back-scatter diffraction (EBSD) was performed to microstructural of HOPG surface, with average grain size 12.09 μm and median grain size 11.47 μm.

### Transient thermal grating (TTG) setup and its measuring principle

For the measurement of thermal transport processes in materials with high thermal conductivity, the transient grating technique is an effective experimental tool[30,31]. Compared to other characterization techniques, TTG offers unique advantages for observing hydrodynamic phonon transport phenomena, such as second sound, particularly at room temperature. Observing second sound requires accessing a specific spatial scale ($MPF_N < L < MPF_R$) and achieving high sensitivity to weak temperature oscillations. While techniques like time- or frequency-domain Thermoreflectance (TDTR/FDTR) are powerful, they are typically limited by modulation frequencies that fall below the hydrodynamic window of graphite or rely on indirect frequency-domain responses that cannot directly capture wave-like thermal transport. Similarly, Brillouin Light Scattering (BLS), though effective for probing single phonon modes, faces challenges in detecting the collective phonon motion characteristic of second sound with weak signal. In contrast, TTG allows flexible control over different transport spatial scales and provides highly sensitive signal detection, enabling direct observation of second sound in the time-domain.

The TTG setup used in our work is shown in Supplementary Fig. 6. A high-power femtosecond laser oscillator (Chameleon Discovery NX, Coherent) is used to generate 80 MHz femtosecond laser pulses with a fixed wavelength laser of 1040 nm, and a tunable laser wavelength range from 660 nm to 1320 nm. A second harmonic generator (HarmoniXX SHG, APE) is used to vary the tunable laser wavelengths with range from 330 nm to 660 nm, from which we select a wavelength of 490 nm. The fixed-wavelength laser is frequency-doubled to produce 520 nm laser by a BBO crystal. The 490 nm laser is modulated by an electro-optic modulator (EOM, M350-160, Conoptics) at 500 kHz and delayed by an optical delay stage up to ~2.5 ns relative to the 520 nm laser. Then, two lasers are non-collinearly focused onto the phase mask through a convex lens. The subsequent convex lens (Lens 1, Lens 2) assembly focuses the ±1 orders of the 490 nm laser onto the sample, which acts as the pump beams.

The absorption of the pump energy by the sample leads to the formation of a spatial sinusoidal temperature distribution, which called transient grating. The period of the transient grating is determined by the angle between the two pump beams. Different phase masks can diffract the incident laser into light beams with different angles, which, after being collected by lens 1 and lens 2, are incident on the sample surface at the corresponding different angles. Thus, the manipulation of the transient grating period can be easily achieved by switching the phase mask. Similarly, the 520 nm laser also diffracts from the same position, one order of its diffracted beams serves as a probe beam, while the other order is used to detect the thermal excitation signal, which is superposed with the diffracted signal of the probe beam to achieve heterodyne detection[32]. The phase plate is used to precisely adjust the phase difference between the probe and the reference, which sets the heterodyne detection mode of TTG to be "amplitude mode" or "phase mode"[30]. In our experiments, phase mode is chosen since graphite has weak thermal reflectance coefficient, hence its thermal expansion is detected instead[5].

The heterodyned detection brings two benefits in TTG. One is the amplification of the typically weak diffraction probe signal; the other is that the detection became phase-sensitive, meaning that the TTG can identify the sinusoidal thermal gratings with different phases, with its output signal proportional to the cosine function value at the grating phase. For examples, zero phase means temperature peak at the pump-interference peak–temperature valley at the pump-interference valley, and π phase means temperature peak at the pump-interference valley–temperature valley at the pump-interference peak. TTG can identify the difference, and output a positive value for zero phase grating, but a negative value for the π phase grating. And the magnitude of the signal is proportional to the temperature difference between peak and valley. In a word, TTG measure the temperature difference between the pump-interference peak position and the pump-interference valley position. If T at the pump interference peak

is larger than T at the valley, TTG signal is positive; if T at the pump interference valley is larger than T at the interference peak, then TTG signal flips to negative.

In our experiment, the single-pulse energies of pump and probe beams are set to 0.24 nJ and 0.12 nJ, respectively, with $1/e^2$ diameters of ~60 µm on the sample surface. The heterodyne-amplified signal beam passes through a sharp long-pass filter to remove the pump background, and is then received by a custom silicon photodetector. The TG signal is measured by a lock-in amplifier synchronized with the EOM. Although the TTG system based on optical delay possesses high time resolution, it is only capable of scanning physical processes within a 2.5 ns duration, limited by the length of the delay stage. For measure the thermal transport processes under larger grating periods, we replace the femtosecond pulsed laser with a CW laser as the probe beam and use a High-bandwidth avalanche photodiode (C5658, Hamamatsu) with high-speed oscilloscope (WP254HD, Lecroy) to probe slower processes of thermal transport.

## Simulation of the normalized thermal expansion decay

Although our TTG setup is in the reflection mode where generally both in-plane and cross-plane thermal transport need to be considered, the transport signal in our current study is essentially 1D case for the following reasons. Graphite has much smaller cross-plane thermal conductivity than its in-plane one. As shown in Supplementary Fig. 7, when the temperature gradient along the cross-plane direction gets smaller shortly after the pump excitation, thermal transport is mainly dictated by the in-plane situation and becomes quasi-one-dimensional for small gratings with micrometer-scale periods[30]. Meanwhile, as the phase-mode TTG measures thermal expansion of graphite integrated along the thickness direction, the cross-plane transport has little effect on the measured signals, and the signal is only sensitive to the in-plane thermal conductivity.

## The second sound excited by TTG

The animation shown in Supplementary Movie 1 demonstrates the picture of second sound at microscopic distances, where the surface undulations of the sample reflect the relative distribution of the temperature difference. The blue dot indicates the actual variation at a specific point (the central point) on the sample, and the corresponding red curve shows the evolution of the temperature difference between the pump-interference peak and valley. When the interference of two pump beams (in our work is 490 nm) focuses on the sample surface to form a transient grating, the sinusoidal distribution energy of the grating induces a sinusoidal thermal grating distribution. At the time of the pump laser energy injection, the surface of the sample gets heated instantaneously, resulting in the strongest relative temperature difference, with the peak shown in the red curve. Subsequently, the heat in the form of the sinusoidal distribution propagates from the peaks to the valleys in a wavelike manner (actually a decaying standing wave). During this process, the red curve shows the relative temperature difference between the pump-interference peak and valley oscillating through several cycles until it reaches its initial state of zero. (This animation exaggerates the effects of the second sound phenomenon for better understanding.)

## TTG signals for different transport regimes

As discussed above, the heterodyned TTG essentially measures the temperature difference between the interference peak position and the interference valley position. And this detection rule will produce different signal shapes under different phonon transport regimes. For examples, in diffusive phonon transport, once the initial sinusoidal thermal grating is excited by the pump laser, its temporal variation is just a production of the sinusoidal pattern with an exponential decay, $\Delta T\cos(qx)\exp(-t/\tau)$, i.e., the temperature difference between peak and valley decays exponentially, which accordingly means that TTG signal

in the diffusive regime is an exponential decay. However, in hydrodynamic phonon transport, due to the thermal wave nature, the temporal evolution of the initial excited thermal grating is a damped standing wave, $\Delta T\cos(qx)\cos(\omega_r t)\exp(-\omega_i t)$, i.e., the temperature difference between peak and valley oscillates from positive to negative and decays exponentially at the same time, which accordingly means that TTG signal in the hydrodynamic regime is a damped oscillation (see Supplementary Movie 1 for an animation of second sound measurement in TTG). Lastly, in ballistic phonon transport, the temporal variation of the initial thermal grating is the result of the superposition of multiple independent phonon modes with different wavevector, group velocity, and polarization. Such complex summation has no analytical form, but typically manifest itself as a random aperiodic oscillation with a positive background in TTG signal[6]. Therefore, the distinction of TTG signals provides a convenient way to spot second sound in phonon hydrodynamic transport. Since both diffusive and ballistic signals are positive and nonperiodic, a damped oscillation, or equivalently a sign flip between positive and negative in a temporally symmetric pattern in the TTG measurement, is the hallmark of second sound phenomenon.

## Details of numerical methods

Our ab initio calculations were performed using the Vienna Ab Initio Simulation Package (VASP)[33–35]. For the exchange and correlation functional, we employed the local density approximation (LDA), along with corresponding projector-augmented-wave (PAW) pseudopotentials[35]. To capture the effects of van der Waals forces, we employed the optB88 functional to explicitly adjust the electron density[36,37]. For structural relaxation, the Brillouin zone was sampled using a Monkhorst-Pack mesh set to $24 \times 24 \times 10$. A real-space supercell method was then used to compute the force constants, second-order constants were derived from a $5 \times 5 \times 2$ supercell with a $6 \times 6 \times 6$ k-point mesh, while third-order constants were calculated using a $4 \times 4 \times 2$ supercell paired with an $8 \times 8 \times 6$ grid. The Phonopy suite[38] was utilized to extract the second-order force constants, and thirdorder.py along with ShengBTE were applied to obtain the third-order constants and the phonon scattering matrix on a $17 \times 17 \times 8$ wavevector grid[39].

Second sound is intrinsically a kinetic manifestation of collective phonon dynamics and therefore requires a microscopic transport description beyond Fourier-type diffusion models. Fractional heat transport models and Cattaneo−Vernotte (CV) equation have been proposed to describe anomalous diffusion in systems where energy carriers exhibit Lévy-type flight statistics or long-range spatiotemporal correlations[40–42].

In such models, nonlocality in space and/or time is introduced phenomenologically through fractional derivatives, leading to non-Gaussian heat spreading and scale-free superdiffusive behavior[43]. In comparison, the Cattaneo−Vernotte (CV) equation extends Fourier's law by introducing a finite relaxation time to correct the assumption of instantaneous heat-flux response, resulting in a hyperbolic heat equation with finite propagation speed. Despite these differences, both approaches remain phenomenological and lack an explicit description of microscopic phonon scattering and collective momentum transport[44,45], and therefore are not suitable for modeling second sound.

In this work, we adopt the phonon Boltzmann transport equation (BTE), which provides a mode-resolved and time-dependent description of phonon populations and naturally unifies diffusive and wave-like heat transport within a single theoretical framework. It has been well established that second sound emerges directly from the BTE when momentum-conserving normal phonon−phonon scattering dominates over resistive processes, without invoking additional phenomenological assumptions[19]. For this reason, the BTE has long been regarded as the minimal and physically grounded

framework for analyzing phonon hydrodynamics and wave-like thermal transport[46,47].

Crucially, an accurate description of second sound requires solving the BTE with the full phonon scattering matrix rather than within the relaxation-time approximation (RTA). While computationally convenient, the RTA fails in the hydrodynamic regime by violating conservation laws and suppressing inter-mode coupling associated with dominant normal scattering, which is the microscopic origin of collective phonon drift and second sound[8]. In contrast, the full scattering matrix formulation rigorously preserves energy and momentum conservation and retains the off-diagonal couplings between phonon modes, enabling a physically correct description of phonon momentum redistribution, collective propagation, and wave damping. Recent theoretical developments, including Green's-function solutions of the linearized BTE, have demonstrated that the full scattering matrix BTE can correctly capture transient thermal gratings and second sound oscillations in materials such as graphite and graphene[19]. As a result, this approach has become the standard and widely accepted theoretical framework for modeling phonon hydrodynamics and second sound in low-dimensional and high-Debye-temperature materials, and it forms the basis of the present theoretical analysis. We now briefly describe the formalism of the full scattering matrix BTE we employed.

Given an arbitrary volumetric heat generation rate $Q(\vec{r}, t)$ in an infinite anisotropic crystal, we wish to calculate the phonon distribution function $f_n(\vec{r}, t)$ and temperature response $T(\vec{r}, t) = T_0 + \Delta T(\vec{r}, t)$, where $T_0$ is the background reference temperature, and $\Delta T$ is the temperature change due to the heating $Q$. Assuming that deviations from the thermal equilibrium distribution are small, the linearized phonon BTE with the full scattering matrix takes the form:

$$\frac{\partial f_n}{\partial t} + \vec{v}_n \cdot \vec{\nabla} f_n = Q_n \frac{N\upsilon}{\hbar \omega_n} + \sum_j W_{n,j}\left(f_j^0 - f_j\right) \quad (1)$$

where $n$ is a short-hand index for a given phonon mode (branch and wavevector in the Brillouin zone), $\omega_n$ is the frequency of the given phonon mode, $f_j$ is the non-equilibrium distribution function, $N$ is the number of discretized points in the Brillouin zone, $\upsilon$ is the unit cell volume and $f_j^0$ is the equilibrium (Bose–Einstein) distribution function. $W_{n,j}$ is the phonon scattering matrix.

We express Eq. (1) using the deviational phonon energy density:

$$\frac{\partial g_n}{\partial t} + \vec{v}_n \cdot \vec{\nabla} g_n = Q p_n + \sum_j \omega_n W_{n,j} \frac{1}{\omega_j}\left(c_j \Delta T - g_j\right) \quad (2)$$

Where $g_n = \frac{\hbar \omega_n}{N\upsilon}\left[f_n - f_n^0(T_0)\right]$ and the volumetric heat generation rate $Q_n$ is replaced by $Q p_n$, where $Q$ is the macroscopic volumetric heat generation rate, and $p_n$ corresponds to how much a given mode is excited by the heating. We assume initial heating is thermally distributed, i.e., $p_n = c_n/C$, where $C = \sum_n c_n$ is the heat capacity.

The BTE given by Eq. (2) describes transport where not only the deviation from the equilibrium distribution at the local temperature is small, but the deviation of the latter from the background constant temperature distribution is also small. The energy density above the background is given simply by $\sum_n g_n$ and the heat flux by $\sum_n g_n \vec{v}_n$. In this linearized regime, the scattering matrix W depends on the background temperature $T_0$ but not on the temperature rise $\Delta T$. Then, the temperature rise as the ratio of the nonequilibrium energy density of phonons divided by the heat capacity:

$$\Delta T = \frac{1}{C}\sum_n g_n \quad (3)$$

To solve for the phonon distribution for a system with no boundaries, we take the spatial and temporal Fourier transform of Eq. (2) to convert the differential equation into an algebraic matrix equation, and find the Fourier transform of the deviational nonequilibrium distribution function in terms of the temperature:

$$\widetilde{g}_n = \widetilde{Q}\sum_j A_{n,j}^{-1} p_j + \Delta\widetilde{T}\left(c_n - i\sum_j A_{n,j}^{-1}\left(\omega + \vec{q}\cdot\vec{v}_j\right)c_j\right) \quad (4)$$

where the matrix A is defined as $A_{n,j} = W_{n,j}\frac{\omega_n}{\omega_j} + i\delta_{n,j}\left(\omega + \vec{q}\cdot\vec{v}_n\right)$, tilde denotes the Fourier transform, $\omega$ represents the temporal frequency from the Fourier transform, not to be confused with the frequency of a phonon mode $\omega_n$, and $\vec{q}$ is the spatial wave vector from the Fourier transform. We find the temperature by inserting Eq. (4) into Eq. (3) and solving to obtain:

$$\Delta\widetilde{T} = \widetilde{Q}\frac{\sum_{n,j} A_{n,j}^{-1} p_j}{i\sum_{n,j} A_{n,j}^{-1} c_j\left(\omega + \vec{q}\cdot\vec{v}_j\right)} \quad (5)$$

Equation (5) represents the frequency-domain response of the temperature field. Consequently, the temporal dynamics of the TTG signal can be obtained by performing an inverse Fourier transform of Eq. (5), which corresponds to the product of the heat source and the Green's function evaluated at the wave vector q associated with the TTG period. More details of this formalism can be found in ref. 19.

Using the aforementioned methodology, we obtain the temperature response of isotope-purified graphite (0.07% C$^{13}$) at 300 K under a sinusoidal excitation with a grating period of 0.95 μm, as shown in Supplementary Fig. 8.

## The effective conductivity $k$ and its compositions in hydrodynamic regime

Extracting the effective thermal conductivity in the transient phonon hydrodynamic regime remains a challenging task. We start from the BTE with Callaway's scattering approximation:

$$\frac{\partial n_\mu}{\partial t} + \boldsymbol{v}_\mu \cdot \nabla n_\mu = -\frac{n_\mu - n_\mu^0(T)}{\tau_\mu^R} - \frac{n_\mu - n_\mu^D(T)}{\tau_\mu^N} \quad (6)$$

The displaced equilibrium is $n_\mu^D = (\exp[\hbar(\omega_\mu - \boldsymbol{q}\cdot\boldsymbol{u})/(k_B T)] - 1)^{-1}$. In the linear regime, we have: $n_\mu^D \approx n_\mu^0(T_0) + C_\mu \Delta T/(\hbar\omega_\mu) + \frac{C_\mu T_0}{\hbar\omega_\mu^2}\boldsymbol{q}\cdot\boldsymbol{u}$, and Eq. (6) is rewritten as:

$$\frac{\partial g_\mu}{\partial t} + \boldsymbol{v}_\mu \cdot \nabla g_\mu = -\frac{g_\mu - C_\mu \Delta T}{\tau_\mu} + \frac{C_\mu T_0}{\omega_\mu \tau_\mu^N}\boldsymbol{q}\cdot\boldsymbol{u}. \quad (7)$$

Using Fourier's transform:

$$\hat{g}_\mu(\omega, \boldsymbol{\xi}) = \int\int g_\mu(t, \boldsymbol{r})\exp(-i\omega t - i\boldsymbol{\xi}\cdot\boldsymbol{r})dt d\boldsymbol{r} \quad (8)$$

We can solve the energy deviation:

$$\hat{g}_\mu = \chi_\mu\left(C_\mu \Delta\hat{T} + \eta_\mu\frac{C_\mu T_0}{\omega_\mu}\boldsymbol{q}\cdot\hat{\boldsymbol{u}}\right). \quad (9)$$

$$\chi_\mu = 1/\left(1 + i\omega\tau_\mu + i\boldsymbol{F}_\mu\cdot\boldsymbol{\xi}\right). \quad (10)$$

where $\eta_\mu = \tau_\mu/\tau_\mu^N$. There are energy and momentum conservation constraints:

$$\sum_\mu \frac{\hat{g}_\mu - C_\mu \Delta \hat{T}}{\tau_\mu} = 0 \tag{11}$$

$$\sum_\mu \hbar \boldsymbol{q} \left( \frac{n_\mu - n_\mu^D}{\tau_\mu^N} \right) = 0 \tag{12}$$

In the Fourier region, we can rewrite Eq. (11–12) as:

$$\begin{bmatrix} \sum_\mu \frac{C_\mu}{\tau_\mu}(1 - \chi_\mu) & \sum_\mu \frac{C_\mu T_0 \boldsymbol{q}^T}{\omega_\mu \tau_\mu^N}(1 - \chi_\mu) \\ \sum_\mu \frac{C_\mu \boldsymbol{q}}{\omega_\mu \tau_\mu^N}(1 - \chi_\mu) & \sum_\mu \frac{C_\mu \boldsymbol{qq}}{\omega_\mu \tau_\mu^N}(1 - \eta_\mu \chi_\mu) \end{bmatrix} \begin{bmatrix} \Delta \hat{T} \\ \hat{\boldsymbol{u}} \end{bmatrix} = \boldsymbol{0} \tag{13}$$

Now we perturbative expand the phonon susceptibility:

$$\chi_\mu = \frac{1}{1 + i\omega\tau_\mu + i\boldsymbol{F}_\mu \cdot \boldsymbol{\xi}} \approx 1 - (i\omega\tau_\mu + i\boldsymbol{F}_\mu \cdot \boldsymbol{\xi}) + (i\omega\tau_\mu + i\boldsymbol{F}_\mu \cdot \boldsymbol{\xi})^2 + \dots \tag{14}$$

Keep to the second order, and perform inverse Fourier transform $(i\omega \to \partial/\partial t, i\boldsymbol{\xi} \to \nabla)$ we can derive the viscous heat equation (VHE):

$$\frac{\partial \theta}{\partial t} + \bar{\tau}\frac{\partial^2 \theta}{\partial t^2} - \frac{\kappa}{C}\frac{\partial^2 \theta}{\partial x^2} + \frac{\Pi}{C}\frac{\partial u}{\partial x} = 0 \tag{15}$$

$$P\frac{\partial u}{\partial t} + \frac{\Pi}{T_0}\frac{\partial \theta}{\partial x} + M\frac{\partial^2 u}{\partial x^2} = -\Gamma u \tag{16}$$

Where $\theta = \Delta T/T_0$ the coefficients are:

$$\bar{\tau} = \sum_\mu C_\mu \tau_\mu / C \tag{17}$$

$$\bar{\tau} = \sum_\mu C_\mu \tau_\mu / C \tag{18}$$

$$\kappa = \sum_\mu C_\mu v_\mu^x F_\mu^x \tag{19}$$

$$\Pi = \sum_\mu \frac{\tau_\mu}{\tau_\mu^N} \frac{C_\mu q^x v_\mu^x}{\omega_\mu} \tag{20}$$

$$M = \sum_\mu \frac{C_\mu}{\omega_\mu^2} (v_\mu^x q^x)^2 \tau_\mu \tag{21}$$

$$P = \sum_\mu \frac{C_\mu (q^x)^2}{\omega_\mu^2} \tag{22}$$

$$\Gamma = \sum_\mu \frac{C_\mu (q^x)^2}{\omega_\mu} \frac{\tau_\mu}{\tau_\mu^N \tau_\mu^R} \tag{23}$$

Simplification of the VHE.

We consider if we can neglect the viscous term in Eq. (9). $\Pi\frac{\partial \theta}{\partial x} + M\frac{\partial^2 u}{\partial x^2}$

$$\Pi\frac{\partial \theta}{\partial x} \sim \Pi\frac{\theta}{L} = \left\langle \frac{C_\mu q^x F_\mu^x}{\omega_\mu \tau_\mu^N} \right\rangle \frac{\theta}{L} \tag{24}$$

$$M\frac{\partial^2 u}{\partial x^2} \sim M\frac{u}{L^2} = \left\langle \frac{C_\mu}{\omega_\mu^2} (v_\mu^x q^x)^2 \tau_\mu \right\rangle \frac{u}{L^2} \tag{25}$$

So:

$$\frac{M\frac{\partial^2 u}{\partial x^2}}{\frac{\Pi}{T_0}\frac{\partial T}{\partial x}} \approx \frac{\left\langle \frac{C_\mu}{\omega_\mu^2}(v_\mu^x q^x)^2 \tau_\mu \right\rangle \frac{u}{L^2}}{\left\langle \frac{C_\mu q^x F_\mu^x}{\omega_\mu \tau_\mu^N} \right\rangle \frac{\Delta T}{T_0 L}} \approx \left\langle \frac{(v_\mu^x)^2 q^x \tau_\mu^N \tau_\mu}{\omega_\mu F_\mu^x} \right\rangle \frac{u}{\theta \cdot L} \tag{26}$$

In strong hydrodynamic region

$$\frac{\Delta T}{\omega_\mu} \sim \frac{T_0 q^x u}{\omega_\mu^2} \tag{27}$$

Therefore, the relation between the two terms are estimated as:

$$\frac{M\frac{\partial^2 u}{\partial x^2}}{\frac{\Pi}{T_0}\frac{\partial T}{\partial x}} \approx \left\langle v_\mu^x \tau_\mu^N \right\rangle \frac{1}{L} \ll 1 \tag{28}$$

Similarly, we consider

$$\frac{\bar{\tau}}{\Delta t} \ll 1 \tag{29}$$

Then we have the following set of PDE:

$$\frac{\partial \theta}{\partial t} - \frac{\kappa}{C}\frac{\partial^2 \theta}{\partial x^2} + \frac{\Pi}{C}\frac{\partial u}{\partial x} = 0 \tag{30}$$

$$P\frac{\partial u}{\partial t} + \Pi\frac{\partial \theta}{\partial x} = -\Gamma u \tag{31}$$

Eliminate one of the variable, we obtained the following PDE (neglecting higher order terms):

$$\frac{\partial^2 \theta}{\partial t^2} + \frac{\Gamma}{P}\frac{\partial \theta}{\partial t} - \left( \frac{\Pi^2 + \Gamma\kappa}{PC} \right)\frac{\partial^2 \theta}{\partial x^2} = 0 \tag{32}$$

Equation (32) is the modified telegraph equation:

$$\frac{\partial^2 \theta}{\partial t^2} + \frac{1}{\tau_{ss}}\frac{\partial \theta}{\partial t} - \frac{D_{eff}}{\tau_{ss}}\frac{\partial^2 \theta}{\partial x^2} = 0 \tag{33}$$

Where $\tau_{ss}$ is the damping time of second sound:

$$\tau_{ss} = \frac{P}{\Gamma} = \frac{\sum_\mu \frac{C_\mu (q^x)^2}{\omega_\mu^2}}{\sum_\mu \frac{\tau_\mu}{\tau_\mu^N} \frac{C_\mu (q^x)^2}{\omega_\mu \tau_\mu^R}} \tag{34}$$

The effective diffusivity is:

$$D_{eff} = \left( \frac{\Pi^2 + \Gamma_\kappa}{PC} \right) \Big/ \left( \frac{\Gamma}{P} \right) = \frac{\Pi^2}{\Gamma} + \frac{\kappa}{C} \tag{35}$$

Thus, the effective conductivity $k$ can be separated to the hydrodynamic contribution and the kinetic contribution:

$$k_{eff} = k_H + k_K \tag{36}$$

Where:

$$k_K = \kappa, \; k_H = \tau_{ss} * v_{ss}^2 * C \tag{37}$$

With the second sound velocity calculated as:

$$v_{ss}^2 = \frac{\Pi^2}{PC} = \frac{\left(\sum_\mu \frac{\tau_\mu}{\tau_\mu^N} \frac{C_\mu q^x v_\mu^x}{\omega_\mu}\right)^2}{\left(\sum_\mu \frac{C_\mu (q^x)^2}{\omega_\mu^2}\right)\left(\sum_\mu C_\mu\right)} \tag{38}$$

### Analysis of TTG signal to obtain second sound speed, propagation length, and the effective $k$

According to the classical Fourier Law of heat conduction, the heat transfer process is driven by the temperature gradient[48]. When we only need to consider heat transfer in one direction (in the x-direction), the one-dimensional Fourier Law can be expressed as Eq. (39):

$$c \frac{\partial \bar{T}}{\partial t} = k \frac{\partial^2 \bar{T}}{\partial x^2} \tag{39}$$

where $c$ is the volumetric heat capacity of the material, $k$ is the thermal conductivity and $\bar{T}$ is the temperature filed.

However, with the small length scales or low temperatures, the unconventional regimes of ballistic phonon transport are observed. In the ballistic regime, the phonons propagate without mutual collisions but only experience boundary scattering. Their effective mean free path is constrained by the material dimensions, leading to a size-dependent thermal conductivity that decreases with reduced system size[21].

Besides ballistic and diffusive phonon transport, hydrodynamic phonon transport, a novel thermal transport behavior was observed experimentally in He[49]. Second sound refers to a peculiar thermal transport phenomenon characterized by wave-like and oscillatory heat propagation, which is indicative of hydrodynamic phonon transport. We use femtosecond transient thermal grating to exited and catch this progress. Due to the transient thermal grating caused by the laser pulse on the sample, the TTG signal can be described by a damped temperature wave equation[6,8]:

$$\frac{\partial^2 \bar{T}}{\partial t^2} + \frac{1}{\tau_{ss}} \frac{\partial \bar{T}}{\partial t} - v_{ss}^2 \frac{\partial^2 \bar{T}}{\partial x^2} = 0 \tag{40}$$

here, $\tau_{ss}$ and $v_{ss}$ are the relaxation time and the velocity of the second sound. This damped wave equation reveal that the second sound evolves in the form of a standing wave under TTG excitation. For a damped temperature wave with grating period $l$, wavevector $k = 2\pi/l$ and frequency $\omega$, the solution of the Eq. (39) can be written as Eq. (40):

$$\Delta T(x, t) = e^{-t/\tau_{ss}} \cos(kx - \omega t) \tag{41}$$

The TTG measured signal $\Delta T$ is proportional to the temperature difference between $\Delta T(0, t)$ and $\Delta T(\frac{l}{2}, t)$:

$$\Delta T(t) = e^{-t/\tau_{ss}} \cos(\omega t) \tag{42}$$

Equation (42) indicates that the second sound, shaped like thermal grating, decays as $\exp(-t/2\tau_{ss})$. Meanwhile, the amplitude of an exponentially decaying wave drops to $exp(-l/2l_{ss})$ over a distance of $l/2$. The position of the first negative peak in the TTG signal represents the time and propagation length required for the second sound to travel half a period. Thus, as shown in Supplementary Fig. 9, the velocity $v_{ss}$ and the propagation length $l_{ss}$ of the thermal waves can be extracted from the dip position $t_d$ and normalized depth $\Delta T_d$ by Eq. (43) and Eq. (44):

$$v_{ss} = \frac{l}{2t_d} \tag{43}$$

$$l_{ss} = \frac{l}{-2\ln(-\Delta T_d)} \tag{44}$$

Therefore, we can extract the speed and the propagation length from the TTG signals, see Fig. 3b, c. At characteristic lengths comparable to phonon mean free paths, particularly within the ballistic and hydrodynamic transport regimes, the assumptions underlying Fourier's law break down. In these regimes, heat transport becomes inherently nonlocal, where the heat flux at a given point depends on the spatiotemporal evolution of the temperature field throughout the entire system rather than a simple local constitutive relation between flux and gradient. Despite the breakdown of the classical diffusive model, employing an "effective" thermal conductivity enables a direct, quantitative comparison with previous literature and provides a unified metric for assessing heat transport efficiency across various transport regimes, from ballistic to hydrodynamic and to diffusive transport.

Following this framework, we derive the effective thermal conductivity by analyzing the temporal decay of the TTG response. Supplementary Fig. 3 presents TTG signals calculated at grating periods of 1, 10, 45, and 300 μm. The black dot lines represent single-exponential fits to the ΔT signals, which are used to extract the thermal conductivity with its damping rate Γ. Furthermore, the effective conductivity $k$ can be expressed as:

$$k = c \frac{\Gamma}{q^2} \tag{45}$$

where $c$ denotes the volumetric heat capacity of graphite, which takes different values at different temperatures[50]. In this equation, the damping rate Γ represents the relaxation of thermal transport. By fitting the time-domain data obtained from simulations, we extract the effective thermal conductivities in Figs. 3d, 4b. The same fitting approach is applied consistently to both simulated and experimental data.

### Reliability of our TTG measurements

Based on the TTG setup, we performed a series of measurements and analyses on HOPG and isotope-enriched graphite. Figure 2 presents the near-room temperature data with clearly signals of second sound in isotope-enriched graphite. With the temperature decreases to 125 K, the differences in thermal transport between HOPG and isotope-enriched graphite become more pronounced across all grating periods, highlighting the strong isotopic effect (Fig. 4a).

Figure 4a displays the low-temperature measurement results of isotope-enriched graphite. These measurements were conducted in a vacuum cryostat (ST-500-UC, Lakeshore) using liquid nitrogen as the cryogen. At low temperatures, the experimental setup becomes increasingly sensitive to temperature fluctuations induced by laser energy injection. In our TTG experiment, each femtosecond laser pulse delivers -35 nJ/mm² to the sample. Simulations performed using COMSOL Multiphysics indicate that the temperature rise due to this energy input is less than 1 K. Given the high repetition rate of the laser pulses (80 MHz) and their low single-pulse energy, as shown in Supplementary Fig. 10a, no significant variation in the second sound signal beyond the noise level was observed when increasing the laser power

density from 20 nJ/mm² to 90 nJ/mm² at 110 K. Further analysis of the equivalent thermal conductivity revealed a decrease of ~4% when comparing the highest and lowest power conditions.

Base temperature of the material is also a critical parameter influencing the TTG signal. We measured the TTG signal of isotope-enriched graphite at a grating period of 4.1 μm over a temperature range of 80 K to 150 K. As shown in Supplementary Fig. 10b, Analysis of the second sound signal revealed an extended propagation length near 125 K, indicating a stronger second sound effect.

For TTG signal acquisition with a high-speed oscilloscope, special attention must be given to whether the system bandwidth is sufficient to capture rapid thermal oscillations accurately. The response of high-bandwidth avalanche photodiode detector is much faster than the heat transport process, ensuring that we have measured the undistorted TTG signal. Supplementary Fig. 10c illustrates the TTG signal measured for HOPG at 300 K with a grating period of 10.4 μm. Notably, when the pump energy is below 25 nJ, the temperature rise induced by laser heating modifies the thermal conductivity by less than 3%.

Supplementary Fig. 11a, b show that TTG signals of second sound are consistently observed across different measurement regions (P1–P6) of the isotope-enriched graphite sample. At both low temperature (100 K) and room temperature (300 K), the experiments reveal a damped oscillation characteristic of second sound. Measurements performed at different regions of the sample further confirm the reliability of observing second sound in isotope-enriched graphite even at room temperature.

## TTG measurement at low temperature

The presence of hydrodynamic transport implies an enhancement in thermal conductivity. Both isotope-enriched graphite (Supplementary Fig. 2a) and HOPG (Supplementary Fig. 2b) exhibit a pronounced second sound effect at 125 K. Supplementary Fig. 2c shows the effective thermal conductivity results for HOPG. By comparing the results of isotope-enriched graphite (Fig. 4a) and HOPG (Supplementary Fig. 2c) at 125 K, it is evident that isotope-enriched graphite has 2 to 3 times larger thermal conductivity than HOPG, suggesting the damping of second sound is significantly reduced in the absence of isotope scattering.

The observed difference between experiment and theory at the bump position of enhanced thermal conductivity is attributed to boundary scattering in graphite grains. At both 300 K (Fig. 3d) and 125 K (Fig. 4a), the experimental results show good agreement with theoretical simulations for small grating periods. However, as the grating period exceeds 10 μm, the effective thermal conductivity gradually deviates from theoretical predictions. For samples with a natural isotope composition, stronger isotope scattering leads to an overall lower effective thermal conductivity compared to isotope-purified samples. Electron backscatter diffraction analysis of HOPG reveals a median grain size of ~10 μm (Supplementary Fig. 4b), which results in significant scattering of long mean free path phonons at low temperatures due to the presence of smaller grains.

Taking the effective thermal conductivity divided by the transport length (grating period), we get the normalized thermal conductivity, which is equivalent to effective thermal conductance. Supplementary Fig. 2d shows the normalized thermal conductivity as a function of grating period. The hydrodynamic peak can still be seen and appears near the transition point into the diffusive regime, indicating again that the hydrodynamic phonons facilitate the transient thermal transport. As the grating period increases sufficiently, thermal transport undergoes a complete transition into the diffusive regime. In this case, the negative linear relation between the logarithmic thermal conductance and the logarithmic transport length can be seen, as thermal conductance $G = kA/L$, $\log_{10}(G) = -\log_{10}(L) + \log_{10}(kA)$.

## Analysis of numerical calculation results of scattering rates

First-principles calculations also reveal distinctions in the number of phonon modes and their diffusivities. In Supplementary Fig. 5a, the horizontal axis represents the number of phonon modes, while the vertical axis represents the corresponding relaxation times. The histogram is constructed with a bin width of 0.01 ns, and the width of each bin indicates the number of resistive-scattering (R) relaxation times falling within the corresponding interval. This statistical representation clearly shows that most of the resistive scattering events occur within a narrow time window (below 0.2 ns), while a smaller fraction of phonon modes exhibits much longer relaxation times. The latter modes can preserve their momentum over extended timescales, serving as the primary carriers of hydrodynamic heat flow.

As shown in Supplementary Fig. 5b, the isotope-scattering rates calculated from first principles further reveal that the resistive scattering channel is significantly weakened in isotope-enriched graphite, where the isotope scattering rate is suppressed by more than an order of magnitude. This suppression substantially enhances the probability for momentum-conserving normal (N) processes to dominate over resistive ones, thereby extending the lifetime of the collective phonon drift. The coexistence of frequent N scattering and reduced R scattering enables the establishment of a quasi-equilibrium phonon distribution characterized by a displaced Bose–Einstein function, a hallmark of phonon hydrodynamics.

These findings demonstrate that the persistence of N scattering and the reduced R damping jointly broaden the temporal and spatial domains where phonon momentum is approximately conserved. As a result, phonon drift can persist over several micrometers and tens of nanoseconds, effectively enhancing the hydrodynamic contribution $k_{\mathrm{H}}$ to the total thermal conductivity $k = k_{\mathrm{K}} + k_{\mathrm{H}}$. The microscopic redistribution of mode populations and their associated scattering times thus provides a solid theoretical foundation for the macroscopic enhancement of the effective thermal conductivity observed in TTG measurements, bridging the Callaway-PBTE framework with experimentally observed phonon hydrodynamic behavior.

## Phonon thermalization of second sound

As shown in Supplementary Fig. 1, the TTG signal can be divided into three stages of dynamical evolution. The electronic peak involves ultrafast carrier excitation, carrier relaxation and electron–phonon coupling; The thermal peak corresponds to intrabranch thermalization of acoustic phonons; and acoustic phonon relaxation associated with second sound after thermal peak. Upon the arrival of the femtosecond pulses (at 0 ps), the sample absorbs photon energy and generates excited carriers with the carrier excitation completed within a few hundred femtoseconds. Subsequently, through electron–phonon coupling, these hot carriers transfer their energy to the TO/LO optical phonons. The optical phonons then relax into acoustic phonons through multiple channels (~2 ps). Meanwhile, the intrabranch thermalization of the nonequilibrium TA/LA phonon populations occurs on a timescale of ~40 ps, which is in good agreement with the findings of Stern et al.[51,52]. This stage is followed by energy redistribution among the acoustic phonon branches with second sound. This thermalization process is indeed important, but its role in the transport mechanism of interest is to facilitate rather than limit hydrodynamic behavior. Hydrodynamic transport emerges whenever the rate of N-scattering exceeds that of U-scattering for the relevant acoustic phonons. From a timescale perspective, as long as N-processes (momentum-conserving scattering) remain sufficiently fast and dominant, the N-process-driven collective motion of phonons can establish a wave-like heat transport even while interbranch thermalization is still ongoing. Microscopically, we believe that once most of the excited electronic energy has been converted to the acoustic phonon system, the second sound should start to move even if the acoustic phonons have not been fully thermalized, because the initially-excited acoustic phonons have well

established the initial momenta. Furthermore, the acoustic phonon thermalization process allows the energy distribution to evolve towards the equilibrium state, thereby forming a more rigorous concept of temperature. Moreover, the transient second sound propagation process will also in turn result in a deviation from the thermalized equilibrium phonon distribution. Thus, more precisely, it is the quasi-temperature (near-thermal, collective phonon energy) difference that is oscillating in our TTG measurement, manifested as the second sound.

## Data availability

All data in the experiments and analysis that support the findings of this study are available. Source data are provided with this study.

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

## Acknowledgements

This work was supported by projects National Key Research and Development Program of China No. 2023YFB4603801; National Natural Science Foundation of China No. 52176173 and No. 21FAA02809; Guangdong Innovative and Entrepreneurial Research Team Program No. 2021ZT09L227; Guang Dong Basic and Applied Basic Research Foundation No. 2020A1515110192, No. 2022A1515010710 and No. 2023B1515040023. The experiments reported were partially conducted at the Guangdong Provincial Key Laboratory of Magnetoelectric Physics and Devices. No. 2022B1212010008.

## Author contributions

Z.D., M.N. and Ke.C. conceived the idea; Z.X., Y.Z., Z.D., D.D., Ku.C., K.W., T.L., T.T., X.Q. and Ke.C. developed the methodology; Z.X., Y.Z., X.H., J.W., X.Q. and Ke.C. conducted the investigations; Z.X., Y.Z., X.H. and Ke.C. prepared the figures; Z.X., Y.Z., X.Q. and Ke.C. wrote the original draft; Z.X., Y.Z., X.H., Z.D., X.Q., M.N. and Ke.C. reviewed and edited the manuscript; T.L., M.N. and Ke.C. supervised the project; T.L. and Ke.C. acquired the funding; Ke.C. was responsible for project administration.

## Competing interests

The authors declare no competing interests.
