## [Transparent Peer Review file · Nature Communications]

Room-temperature second sound in isotopically pure graphite

Corresponding Author: Professor Ke Chen

Version 0:

Reviewer comments:

Reviewer #1

(Remarks to the Author)

The work of Xie et al. entitled 'Room-temperature second sound in isotopically pure graphite' provides evidence of second sound at room temperature using transient grating measurements and discusses the contribution of collective phonon dynamics and phonon hydrodynamics in heat conductivity. These phenomena are enabled by the high isotopic purity of the graphite samples.

The noteworthy results are (1) the detection of a known experimental signature of second sound at room temperature and (2) the investigation of the role of phonon hydrodynamics in heat conductivity with transient grating measurements and theoretical calculations. The findings of the work are interesting and can drive further developments in the field of second sound and phonon hydrodynamics in solids. The results and calculations on heat conductivity can be interesting for thermal engineering. This work is based on previous studies on the detection of second sound at elevated temperatures with transient grating measurements - see Huberman et al. *Science* 364, 375–379 (2019) and Ding et al. *Nat. Commun.* 13, 285 (2022) - but the present work is original. The experimental signature of second sound in transient grating measurements has been extensively discussed in previous works and the calculations based on the phonon Peierls-Boltzmann transport equation are an appropriate method for this problem. The existence of second sound at room temperature in isotopically pure graphite has been previously predicted with ab initio calculations in the work of Ding et al. The isotopic purity is shown with secondary ion mass spectroscopy and Raman measurements (Fig 1b), while its importance is demonstrated experimentally (Figures 2a-d) and theoretically (Fig 3a). The association of the measured transient grating signal with second sound is strengthened by additional measurements showing that the signal is enhanced at lower temperatures, and with comparisons of isotopically pure graphite and HOPG samples.

Apart from these positive aspects I would like to ask the Authors few questions:

1) Something general: The use of the term temperature (which is oscillating in this case) requires a thermal distribution for phonons. According to the work of Stern et al. (PRB, 97(16), 165416, 2018) the thermalization of the ZA branch and the LA/TA interband relaxation can require ~100 ps, which is comparable to the timescales probed here. Some authors from this work have also explored the capability of their technique to probe second sound; see *Struct. Dyn.* 11, 024101 (2024). I wonder how / if this thermalization timescale of acoustic phonons in graphite is relevant for the present work. The Authors already explain the differences between diffusive phonons (exponential decay), ballistic phonons (non periodic oscillations with positive background) and hydrodynamic phonons (damped oscillation) but can they also briefly comment on the aspect of thermalization?

2) The observation of second sound at room temperature is an unexpected and interesting finding. It is possible that this observation is enabled by the high sensitivity of the experimental probe. Regarding independent confirmation, can the Authors briefly comment whether it should be possible to observe RT second sound in this system with relevant pump probe techniques, frequency domain thermoreflectance or Brillouin spectroscopy?

3) The Authors mention that only a handful of solids have shown second sound. This study is focused on graphite that is known to have various interesting properties like high in-plane heat conductivity. But is this material selected here because it can be efficiently prepared in isotopically pure form? The previous works of Huberman et al. and Ding et al. have also studied graphite. Can the Authors extend the motivation for targeting graphite?

4) Fitting the TTG signal with an exponential decay gives the attenuation rate of peak-valley temperature difference (extended data Figure 6), which is then used to extract an effective thermal conductivity. But the TTG signal is not simply an exponential decay due to second sound. Can the Authors explain a bit more about it? Is this what makes the extracted value 'effective'?

Reviewer #2

(Remarks to the Author)

The manuscript reports on sub-nanosecond TTG measurements performed on isotopically purified graphite and on the detection of a wave-like heat-transport regime (second sound) at room temperature. The experimental part is well executed and potentially important, as it raises the question of the conditions under which second sound can be detected in graphite and related 2D materials not only at low temperatures (100–200 K), but also at room temperature. This is the central question and the new scientific contribution of the manuscript.

The theoretical analysis, however, remains confusing. The authors use the Boltzmann transport equation, the Callaway model, ab initio calculations, the misleading terminology of diffusive and hydrodynamic phonons, and introduce the notion of “superdiffusivity” as an increase in thermal conductivity. In contrast, the concept of superdiffusion does exist, but it requires fractional transport models, which are neither mentioned nor considered in the manuscript. Moreover, in its present formulation, the theoretical section does not provide real insight into phonon dynamics in 2D materials, despite the authors’ claims in the conclusion.

The fast TTG dynamics that the authors interpret as “superdiffusive conductivity” are misinterpreted. Such an effect may originate from the wave-like regime (second sound), but its physical origin is not an increase in the thermal conductivity k . Rather, it is due to the delay of the heat flux relative to the temperature gradient—the fundamental distinction between wave-like and diffusive transport, which the authors do not recognize. Although they formally separate the diffusive and wave regimes, the manuscript does not demonstrate a correct understanding of the macroscopic or microscopic origin of their difference.

The additional ab initio calculations and microscopic models, although presented as providing “deeper insight” into phonon dynamics, do not in fact offer such insight: the models are overly simplified, and the cited literature omits key theoretical works on phonons, phonon dynamics in 2D materials, wave-like heat transport, relaxation times, and fractional transport theories relevant to superdiffusion. As a result, the theoretical section is extensive but does not meaningfully contribute to understanding the experiment or the state of the field.

The only real contribution of the manuscript—potentially significant—lies in explaining why the authors detect a wave-like effect at room temperature in graphite, whereas previous TTG studies on graphene (Huberman 2019, Ding 2022) did not. To achieve this, the authors would need to clearly discuss differences in the material itself, isotopic purity, sample thickness, accessible grating-period range, and temporal resolution of the measurements. It is also essential that they provide a quantitative comparison of the obtained heat-propagation speed and thermal relaxation time or damping with previous works, to clarify what is truly new in their experiment and what enables the observation of second sound at room temperature.

In its current form, the manuscript combines a solid experimental result with a theoretical narrative that is overly ambitious, imprecise, often conceptually incorrect, and misguided—substantially reducing its scientific value and focus.

I suggest that authors consider withdrawing the current manuscript and preparing two clearly focused papers:

- One dedicated to the detection of second sound with a detailed comparison to previous studies.
- Another dedicated to the analysis of the TTG method for determining thermal conductivity, without the introduction of inaccurate microscopic models and the problematic use of the “superdiffusivity” concept. For deeper insight into phonon processes, use physically consistent theoretical approaches to phonon dynamics in 2D materials and their connection to macroscopically measurable parameters. For superdiffusion, use fractional heat-transport models, not conceptually incorrect categorical distinctions.

I also suggest the authors consult the following relevant references:

DOI: 10.1103/RevModPhys.61.41

DOI: 10.1103/PhysRevB.87.115421

[https://doi.org/10.1016/S1369-7021\(12\)70117-7](https://doi.org/10.1016/S1369-7021(12)70117-7)

<http://dx.doi.org/10.1016/j.spmi.2015.09.027>

<https://doi.org/10.1038/s41467-019-11572-4>

DOI: 10.1103/PhysRevB.107.054311

DOI: 10.1088/0305-4470/30/21/006

Mainardi, F. *Fractional Calculus and Waves in Linear Viscoelasticity*, Imperial College Press, 2010

<https://doi.org/10.3390/fractalfract9100653>

<https://doi.org/10.3389/fphy.2017.00052>

Reviewer #3

(Remarks to the Author)

This is an impressive experimental achievement. The authors observed second sound at room temperature in graphite, a result considered nearly impossible due to the extreme sensitivity of second sound to the isotopic purity of the material. The

success was made possible by the preparation of a highly isotopically pure sample.

This work is fundamentally important because it demonstrates that phonon hydrodynamics can be accessed and exploited at room temperature. The observation of second sound in this context is a significant step forward, as it opens the door to practical applications involving materials with high thermal conductivity. Given these advancements, this work is certainly deserving of publication.

Version 1:

Reviewer comments:

Reviewer #1

(Remarks to the Author)

The Authors provided detailed responses to all my comments. Their responses were accompanied by appropriate revisions of the manuscript and inclusion of additional extended data that I found very informative. I also appreciate their detailed explanations regarding some general questions I had.

At this stage, I do not have additional comments or questions.

The observation of second sound at room temperature - which is the main experimental finding of this work - is interesting and deserves publication. The Authors have provided enough data and a detailed discussion of their experimental and theoretical methods so that others can reproduce or extend their work. I believe that this article will attract attention from the community and trigger further developments. Therefore, I recommend publication.

Reviewer #2

(Remarks to the Author)

The manuscript presents a very strong experimental contribution with original results that are significant for the field. The revised version substantially improves the explanation of the experimental setup. It is now clearly shown that the authors, thanks to the improved temporal and spatial resolution of the TTG setup and the use of isotopically purified graphite, were able to detect the oscillatory signal interpreted as second sound at room temperature. This part of the work is very strong and scientifically valuable.

The authors have also addressed the first-round comment by removing the term super-diffusive, avoiding serious terminological confusion that could have misled readers regarding the model used.

However, despite these significant improvements, there remains a fundamental issue in the manuscript's structure and interpretation that requires clarification.

1. Role of ab initio and BTE in relation to the experimental analysis

From the manuscript structure, it appears that the ab initio / BTE results are not directly used. The workflow can be summarized as follows:

- ab initio and linearized BTE are used to obtain the temperature response in the Fourier domain (Eq. 1–5);
- through further simplifications and the Callaway approximation, a system is obtained that reduces to a modified telegraph (Cattaneo–Vernotte type) equation (Eq. 33);
- Eq. (33) is then used to interpret the TTG signal and fit the oscillatory behavior;
- The effective conductivity k_{eff} is determined by additional empirical fitting of the exponential decay of the signal via $k=c\Gamma/q^2$.

From this procedure, it is clear that in the analysis of the experimental data (Figs. 1–4), the results of ab initio / BTE calculations are not directly used; rather, a reduced macroscopic model and parameters obtained from fitting the TTG signal are employed. However, the manuscript does not make this distinction sufficiently clear, leaving the quantitative role of the ab initio / BTE part in the experimental interpretation ambiguous.

2. Interpretation of the results in Figures 3 and 4

The results in Figures 3 and 4 are very interesting, but their interpretation through the decomposition $k_{\text{eff}}=k_K+k_H$ and the division into “hydrodynamic”, “ballistic”, and “diffusive” contributions gives the impression that the experiment directly separates these contributions. According to the methodology, k_{eff} is derived from the time decay of the TTG signal, while its decomposition relies on an additional theoretical construction not directly linked to the fitting procedure or to the ab initio / BTE part.

3. Terminology issues

The use of terms such as hydrodynamic phonons, ballistic phonons, diffusive phonons, and collective phonons may lead to confusion. In standard terminology, these terms refer to transport regimes rather than phonon types. In the manuscript, they are used in a way that suggests a different classification, which requires explicit clarification—especially given their impact on the interpretation of the results in Figures 3 and 4.

Key concern

The way the theoretical part is connected to the experimental analysis could lead readers to incorrect conclusions about what is actually measured and what represents the theoretical framework for interpretation. This appears primarily as a structural and terminological issue rather than a problem with the results themselves.

Recommendation for clarification (without changing the results)

1. Clearly separate the experimental analysis based on Eq. (33) from the theoretical context based on ab initio / BTE calculations.
2. Precisely explain how κ_{eff} is obtained and interpreted.
3. Provide clearer definitions for the terminology used regarding transport regimes and phonons.

Conclusion

The work presents a strong experimental contribution, but a major revision is necessary to clarify the narrative, terminology, and the relation between the theoretical and experimental parts of the manuscript.

Reviewer #3

(Remarks to the Author)

I believe that the authors addressed all Referee's remarks, so the manuscript is ready for publication,

Version 2:

Reviewer comments:

Reviewer #2

(Remarks to the Author)

The experimental results presented in this work are convincing and represent a valuable contribution to the study of second sound in graphite, particularly in demonstrating the role of isotope purification in enabling its detection at room temperature. I would like to emphasize, however, that the interpretation of the results is strongly framed within the commonly used hydrodynamic/ballistic/diffusive phonon picture based on simplified Callaway-type PBTE arguments. While widespread in current literature, this framework offers a heuristic and highly reduced description of heat transport and does not fully reflect the broader theoretical context of wave-like thermal transport developed in statistical physics and non-equilibrium thermodynamics.

In my view, some of the claims regarding "enhancement of effective thermal conductivity beyond the diffusive limit" arise from this particular interpretative framework rather than being a direct experimental observable.

That said, the experimental observations themselves remain solid and important. The interpretation presented here should therefore be understood as one possible theoretical perspective, for which the authors take responsibility.

Authors' Response to Reviewer Comments

We sincerely thank the Reviewers for their thoughtful and constructive comments. Their valuable feedback has been instrumental in enhancing the clarity, depth, and overall quality of the manuscript. In response, we have added more discussion of the experiments, refined the theoretical analysis, and thoroughly revised the manuscript. These improvements significantly enhance the overall quality of the work. Below, we provide detailed, point-by-point responses to all reviewers' comments (reviewer comments are presented in **blue**, our response is presented in **black**).

Reviewer #1:

The work of Xie et al. entitled 'Room-temperature second sound in isotopically pure graphite' provides evidence of second sound at room temperature using transient grating measurements and discusses the contribution of collective phonon dynamics and phonon hydrodynamics in heat conductivity. These phenomena are enabled by the high isotopic purity of the graphite samples.

The noteworthy results are (1) the detection of a known experimental signature of second sound at room temperature and (2) the investigation of the role of phonon hydrodynamics in heat conductivity with transient grating measurements and theoretical calculations. The findings of the work are interesting and can drive further developments in the field of second sound and phonon hydrodynamics in solids. The results and calculations on heat conductivity can be interesting for thermal engineering. This work is based on previous studies on the detection of second sound at elevated temperatures with transient grating measurements - see Huberman et al. *Science* 364, 375–379 (2019) and Ding et al. *Nat. Commun.* 13, 285 (2022) - but the present work is original. The experimental signature of second sound in transient grating measurements has been extensively discussed in previous works and the calculations based on the phonon Peierls-Boltzmann transport equation are an appropriate method for this problem. The existence of second sound at room temperature in isotopically pure graphite has been previously predicted with ab initio calculations in the work of Ding et al. The isotopic purity is shown with secondary ion mass spectroscopy and Raman measurements (Fig 1b), while its importance is demonstrated experimentally (Figures 2a-d) and theoretically (Fig 3a). The association of the measured transient grating signal with second sound is strengthened by additional measurements showing that the signal is enhanced at lower temperatures, and with comparisons of isotopically pure graphite and HOPG samples.

Response: We thank the reviewer for the recognition of the importance of our work, as well as the valuable comments and suggestions. We appreciate the positive evaluation of our experimental evidence for room-temperature second sound in isotopically pure graphite. Following the Reviewer's comments, we have revised the manuscript to

improve clarity and completeness, as detailed in the responses below.

Apart from these positive aspects I would like to ask the Authors few questions:

1) Something general: The use of the term temperature (which is oscillating in this case) requires a thermal distribution for phonons. According to the work of Stern et al. (PRB, 97(16), 165416, 2018) the thermalization of the ZA branch and the LA/TA interband relaxation can require ~ 100 ps, which is comparable to the timescales probed here. Some authors from this work have also explored the capability of their technique to probe second sound; see Struct. Dyn. 11, 024101 (2024). I wonder how / if this thermalization timescale of acoustic phonons in graphite is relevant for the present work. The Authors already explain the differences between diffusive phonons (exponential decay), ballistic phonons (non periodic oscillations with positive background) and hydrodynamic phonons (damped oscillation) but can they also briefly comment on the aspect of thermalization?

Response: We thank the reviewer for pointing out the issue of phonon thermalization. This is a very insightful and crucial question, as it addresses the microscopic foundation of heat transport phenomena. To address the reviewer's concerns regarding phonon thermalization, we have added more discussion in the Extended Data Fig. 1. (Page 29, Lines 919 - 945) of the revised manuscript.

In our experiment, as demonstrated by the Extended Date Fig. 1, the intrabranh thermalization process of the non-equilibrium TA/LA phonon populations following laser excitation takes approximately 40 ps. This is in good agreement with the findings of Stern et al¹. This stage is immediately followed by the energy redistribution between acoustic phonon modes (the relaxation between the LA/TA) with second sound².

This thermalization process is important, yet its role in the transport mechanism of interest is to facilitate rather than limit hydrodynamics. We maintain that hydrodynamic transport occurs provided the condition Rate (N-scattering $>$ U-scattering) is met for the relevant phonon modes. From a timescale perspective, as long as N-processes (momentum conservation) are sufficiently fast and dominant, the driven by N-process transport will be established and the collective phonon will begin to motion like wave, even while interband thermalization is still ongoing. Microscopically, we believe that once most of the excited electronic energy has been converted to the acoustic phonon system, the second sound should start to move even if the acoustic phonons have not been fully thermalized, because the initially-excited acoustic phonons have well established the initial momenta. Furthermore, the acoustic phonon thermalization process allows the energy distribution to evolve towards the equilibrium state, thereby forming a more rigorous concept of temperature. Moreover, the transient second sound propagation process will in turn result in a deviation from the thermalized equilibrium phonon distribution. Thus, more precisely, it is the quasi-temperature (near-thermal,

collective phonon energy) difference that is oscillating in our TTG measurement, manifested as the second sound.

Revisions to the manuscript:

On Results and discussion (Page 6, Lines 180 - 182):

“TTG essentially probes the difference in quasi-temperature (i.e., the collective phonon energy) between the interference peak and valley positions, while remaining generally unaffected by the thermalization of the acoustic phonons (see Methods and Extended Data Fig. 1 for details).”

On Methods — Phonon thermalization of second sound (Page 29, Lines 918 - 945):

“As shown in Extended Data Fig. 1, the TTG signal can be divided into three stages of dynamical evolution. The electronic peak involves ultrafast carrier excitation, carrier relaxation and electron–phonon coupling; The thermal peak corresponds to intrabranh thermalization of acoustic phonons; and acoustic phonon relaxation associated with second sound after thermal peak. Upon the arrival of the femtosecond pulses (at 0 ps), the sample absorbs photon energy and generates excited carriers with the carrier excitation completed within a few hundred femtoseconds. Subsequently, through electron–phonon coupling, these hot carriers transfer their energy to the TO/LO optical phonons. The optical phonons then relax into acoustic phonons through multiple channels (~2 ps). Meanwhile, the intrabranh thermalization of the nonequilibrium TA/LA phonon populations occurs on a timescale of approximately 40 ps, which is in good agreement with the findings of Stern et al. This stage is followed by energy redistribution among the acoustic phonon branches with second sound. This thermalization process is indeed important, but its role in the transport mechanism of interest is to facilitate rather than limit hydrodynamic behavior. Hydrodynamic transport emerges whenever the rate of N-scattering exceeds that of U-scattering for the relevant acoustic phonons. From a timescale perspective, as long as N-processes (momentum-conserving scattering) remain sufficiently fast and dominant, the N-process-driven collective motion of phonons can establish a wave-like heat transport even while interbranch thermalization is still ongoing. Microscopically, we believe that once most of the excited electronic energy has been converted to the acoustic phonon system, the second sound should start to move even if the acoustic phonons have not been fully thermalized, because the initially-excited acoustic phonons have well established the initial momenta. Furthermore, the acoustic phonon thermalization process allows the energy distribution to evolve towards the equilibrium state, thereby forming a more rigorous concept of temperature. Moreover, the transient second sound propagation process will also in turn result in a deviation from the thermalized equilibrium phonon distribution. Thus, more precisely, it is the quasi-temperature (near-thermal, collective phonon energy) difference that is oscillating in our TTG measurement, manifested as the second sound.”

On the References:

[Ref. 48] Stern, M. J. *et al.* Mapping momentum-dependent electron-phonon coupling and nonequilibrium phonon dynamics with ultrafast electron diffuse scattering. *Phys. Rev. B* **97**, 165416 (2018).

[Ref. 49] Kremeyer, L., Britt, T. L., Siwick, B. J. & Huberman, S. C. Ultrafast electron diffuse scattering as a tool for studying phonon transport: Phonon hydrodynamics and second sound oscillations. *Struct. Dyn.* **11**, 024101 (2024).

On Extended Data Fig. 1:

Extended Data Fig. 1. The TTG Signal of whole dynamical process for isotope-purified graphite at 300 K with 0.95 μm grating period. The TTG signal data illustrate the complete dynamical process following laser excitation. The data can be broadly divided into three stages of dynamical evolution. The electronic peak involves ultrafast carrier excitation, carrier relaxation and electron-phonon coupling (up to the first vertical dashed line, ~ 2 ps). The thermal peak corresponds to intrabrand thermalization of acoustic phonons (up to the second vertical dashed line, ~ 40 ps). After thermal peak, acoustic phonons relax with second sound.

2) The observation of second sound at room temperature is an unexpected and interesting finding. It is possible that this observation is enabled by the high sensitivity of the experimental probe. Regarding independent confirmation, can the Authors briefly comment whether it should be possible to observe RT second sound in this system with relevant pump probe techniques, frequency domain thermorefectance or Brillouin spectroscopy?

Response: We appreciate the reviewer's suggestion regarding independent verification of second sound with other experimental techniques. From theoretical and experimental perspectives, observing second sound requires satisfying two conditions: accessing the

appropriate spatial scale for hydrodynamic phonon transport ($MPF_N < L < MPF_R$), and achieving sufficient sensitivity to detect small temperature differences.

Transient Thermal Grating (TTG):

TTG provides flexible control over the transport spatial scale via adjustable grating periods. When combined with pump–probe methods, it achieves high temporal resolution and sensitive detection of quasi-temperature variations. TTG therefore remains the method of choice for studying second sound in solids, offering a clear and direct physical picture.

Time-Domain Thermoreflectance (TDTR):

TDTR can access similar temporal and spatial scales. However, it is typically limited by laser and modulation systems, operating at 1–10 MHz, which is below the hydrodynamic window relevant for second sound in graphite.

Frequency-Domain Thermoreflectance (FDTR):

FDTR provides partial information on thermal transport in the frequency domain. For example, Beardo et al. observed second sound in Ge³. Nevertheless, detecting second sound in graphite at room temperature requires extremely high modulation frequencies that exceed current FDTR capabilities. Besides, FDTR probes second sound indirectly through frequency response and cannot directly capture the wavelike thermal transport.

Brillouin Light Scattering (BLS):

BLS is an effective technique for probing phonon dynamics, as it measures frequency shifts arising from light–phonon interactions. However, second sound is fundamentally different: it does not correspond to a single phonon mode but rather to a collective motion of phonons, which is more directly observed in the time domain. Moreover, the second-sound signal at room temperature is extremely weak, demanding high spectral resolution and signal-to-noise ratio, which limits the feasibility of BLS for its detection.

Other pump–probe techniques:

We are also exploring methods such as pump-probe transient reflectivity microscopy using CMOS imaging to directly observe heat propagation⁴. However, acquiring a 2D array signal reduces the signal-to-noise ratio, making the detection of weak second-sound signals extremely challenging.

In summary, while other techniques may provide partial information in principle, TTG currently offers clear advantages for observing room-temperature second sound in graphite. In response to the reviewer’s comment, we have added a discussion of similar experimental techniques in the Methods section.

Revisions to the manuscript:

On Methods — Transient thermal grating (TTG) setup and its measuring principle (Pages 17-18, Lines 513 - 524):

“For the measurement of thermal transport processes in materials with high thermal conductivity, the transient grating technique is an effective experimental tool. Compared to other characterization techniques, TTG offers unique advantages for observing hydrodynamic phonon transport phenomena, such as second sound, particularly at room temperature. Observing second sound requires accessing a specific spatial scale ($MPF_N < L < MPF_R$) and achieving high sensitivity to weak temperature oscillations. While techniques like time- or frequency-domain Thermoreflectance (TDTR/FDTR) are powerful, they are typically limited by modulation frequencies that fall below the hydrodynamic window of graphite or rely on indirect frequency-domain responses that cannot directly capture wave-like thermal transport. Similarly, Brillouin Light Scattering (BLS), though effective for probing single phonon modes, faces challenges in detecting the collective phonon motion characteristic of second sound with weak signal. In contrast, TTG allows flexible control over different transport spatial scales and provides highly sensitive signal detection, enabling direct observation of second sound in the time-domain.”

3) The Authors mention that only a handful of solids have shown second sound. This study is focused on graphite that is known to have various interesting properties like high in-plane heat conductivity. But is this material selected here because it can be efficiently prepared in isotopically pure form? The previous works of Huberman et al. and Ding et al. have also studied graphite. Can the Authors extend the motivation for targeting graphite?

Response: We thank the reviewer for this insightful comment. To date, only a limited number of materials have shown second sound. This fascinating phenomena of heat transport requires phonon–phonon scattering to be dominated by N-processes, with minimal U-scattering.

As noted by the reviewer, graphite’s high in-plane thermal conductivity inherently reflects weak resistive (R) scattering, making it one of the few materials where second sound can theoretically persist to relatively high temperatures. Previous studies by Huberman et al. and Ding et al. have already observed second sound in graphite up to approximately 200 K. Theoretical calculations indicate that graphite has the potential for observing second sound at room temperature through isotopic purification^{5,6}.

Building on these works, we focus on graphite not only for its intrinsic material properties but also because isotopically enriched samples can be prepared. As anticipated by the reviewer, high-quality, large-area, isotopically purified graphite can be reliably synthesized technically, which substantially reduces isotope scattering and facilitates the extension of the hydrodynamic transport regime toward room

temperature.

We are also actively exploring and searching for other materials that can exhibit second sound. The insights gained from graphite are expected to provide important guidance for future studies of thermal transport mechanisms in diamond, two-dimensional materials, and related systems.

Revisions to the manuscript:

On Results and discussion (Page 9, Lines 227 - 231):

“In a word, room temperature drifting second sound is observed in isotope-enriched graphite with TTG measurement, and the results can be well supported by first-principles calculation, demonstrating that the technologically-feasible isotope engineering in graphite (and potentially other materials) indeed opens a new avenue to explore the hydrodynamic wave-like heat transport at room-temperature.”

4) Fitting the TTG signal with an exponential decay gives the attenuation rate of peak-valley temperature difference (extended data Figure 3), which is then used to extract an effective thermal conductivity. But the TTG signal is not simply an exponential decay due to second sound. Can the Authors explain a bit more about it? Is this what makes the extracted value 'effective'?

Response: We thank the reviewer for raising this important point regarding the TTG signal fitting. In the presence of second sound, the TTG response does not follow a simple exponential decay because the signal contains both diffusive and hydrodynamic components. For the second sound, heat transport is better described as a damped thermal wave rather than diffusive relaxation.

In our analysis, we fit the TTG signal with an exponential function to extract a characteristic decay rate. This fitting captures the overall dissipation of thermal energy, rather than the instantaneous oscillations of the temperature wave, providing a well-defined, experimentally accessible measure. This approach allows us to extract a decay rate that reflects the heat transport, even though it does not fully resolve the detailed wave dynamics of second sound.

Consequently, the thermal conductivity should be regarded as an effective, rather than an intrinsic, material property. The concept of thermal conductivity is rooted in Fourier’s law, which assumes a fixed relationship between heat flux and temperature gradient. However, at characteristic length comparable to phonon mean free paths and in the ballistic and hydrodynamic transport regimes, the assumptions underlying Fourier’s law break down. Heat transport becomes inherently nonlocal, meaning that the heat flux at a given point depends not only on the local temperature gradient but also on the temperature distribution throughout the system.

Therefore, the quantity we extract represents an effective thermal conductivity rather than an intrinsic one. Despite the breakdown of Fourier's law in micro- and nanoscale, an effective thermal conductivity remains a widely used and physically intuitive quantity in thermal transport studies, particularly in transient grating experiments⁷. It enables direct, quantitative comparison with previous literature and provides a unified metric for assessing heat transport efficiency across different transport regimes, from ballistic to hydrodynamic and to diffusive transport regimes. For these reasons, we follow an exponential fitting and refer to the extracted quantity as an “effective thermal conductivity” in this work.

Revisions to the manuscript:

On Methods — Analysis of TTG Signal to obtain second sound speed, propagation length, and the effective k (Page 26, Lines 809 - 819):

“At characteristic lengths comparable to phonon mean free paths, particularly within the ballistic and hydrodynamic transport regimes, the assumptions underlying Fourier's law break down. In these regimes, heat transport becomes inherently nonlocal, where the heat flux at a given point depends on the spatiotemporal evolution of the temperature field throughout the entire system rather than a simple local constitutive relation between flux and gradient. Despite the breakdown of the classical diffusive model, employing an “effective” thermal conductivity enables a direct, quantitative comparison with previous literature and provides a unified metric for assessing heat transport efficiency across various transport regimes, from ballistic to hydrodynamic and to diffusive transport. Following this framework, we derive the effective thermal conductivity by analyzing the temporal decay of the TTG response.”

Reviewer #2:

The manuscript reports on sub-nanosecond TTG measurements performed on isotopically purified graphite and on the detection of a wave-like heat-transport regime (second sound) at room temperature. The experimental part is well executed and potentially important, as it raises the question of the conditions under which second sound can be detected in graphite and related 2D materials not only at low temperatures (100–200 K), but also at room temperature. This is the central question and the new scientific contribution of the manuscript.

Response: Thank you very much for the meticulous review and the deep, insightful comments on our manuscript. We are highly appreciative that the reviewer recognized the quality and potential significance of our experimental work and acknowledged the detection of room-temperature wave-like heat transport (Second Sound) as the central and novel scientific contribution.

The theoretical analysis, however, remains confusing. The authors use the Boltzmann transport equation, the Callaway model, *ab initio* calculations, the misleading terminology of diffusive and hydrodynamic phonons, and introduce the notion of “superdiffusivity” as an increase in thermal conductivity. In contrast, the concept of superdiffusion does exist, but it requires fractional transport models, which are neither mentioned nor considered in the manuscript. Moreover, in its present formulation, the theoretical section does not provide real insight into phonon dynamics in 2D materials, despite the authors’ claims in the conclusion.

Response: We have carefully considered the serious criticisms regarding the conceptual confusion and inaccuracy in the theoretical analysis. We have committed to a complete and thorough revision of the theoretical analysis section based on your feedback. Our primary goal in this revision is to correct the flawed concepts, clearly articulate that *Second Sound* is a hydrodynamic phenomenon, and establish a physically sound theoretical framework rooted in the phonon hydrodynamics model to support our experimental findings.

The fast TTG dynamics that the authors interpret as “superdiffusive conductivity” are misinterpreted. Such an effect may originate from the wave-like regime (second sound), but its physical origin is not an increase in the thermal conductivity k . Rather, it is due to the delay of the heat flux relative to the temperature gradient—the fundamental distinction between wave-like and diffusive transport, which the authors do not recognize. Although they formally separate the diffusive and wave regimes, the manuscript does not demonstrate a correct understanding of the macroscopic or microscopic origin of their difference.

Response: We thank the referee for raising important questions regarding the

theoretical framework used to model the wave-like heat transport observed in our TTG experiments. In our study, the fast TTG dynamics are not attributed to an anomalous increase of the thermal conductivity nor to Lévy-type fractional transport, which constitute the physical basis of true superdiffusion. Instead, the effect we observe corresponds to a wave-like heat-transport regime, i.e., second sound, arising from the dominance of normal (N) phonon–phonon scattering over resistive (R) scattering. This is the established criterion for hydrodynamic transport and has been emphasized since Gurzhi (1964) and more recently by G. Chen’s phonon-fluid formulation.

We emphasize that our choice of the phonon *Boltzmann transport equation* (BTE) is motivated by the fact that second sound is fundamentally a kinetic phenomenon arising from the collective dynamics of phonons, which lies outside the validity of Fourier-conduction-based or phenomenological macroscopic transport equations. In contrast to Fourier diffusion equations that assume local equilibrium and instantaneous proportionality between heat flux and temperature gradient, the BTE provides a microscopic, mode-resolved description of phonon populations and their spatiotemporal evolution, and is therefore a widely used framework for capturing both diffusive and wave-like heat transport within a unified formalism. As established in classic and modern literature, second sound emerges naturally from the BTE when normal phonon–phonon scattering dominates over resistive processes, without the need to invoke additional phenomenological assumptions.

Beyond the choice of the BTE itself, we stress that the use of the *full scattering matrix* is essential for a physically correct description of second sound. The relaxation-time approximation (RTA), while convenient, is known to break down precisely in the regime relevant to hydrodynamic phonon transport: it violates energy conservation for mode-dependent relaxation times and artificially suppresses collective phonon behavior^{8,9}. In particular, the RTA cannot correctly capture the coupling between phonon modes induced by dominant normal scattering, which is the microscopic origin of second sound¹⁰. By contrast, the full scattering matrix formulation rigorously preserves conservation laws and retains the off-diagonal mode couplings required to describe phonon momentum redistribution and collective propagation¹¹. As a result, it is capable of predicting both the existence and the damping of second sound waves in materials with high Debye temperatures, such as graphite and graphene.

Importantly, the full scattering matrix BTE has become the *standard theoretical framework* for modeling second sound and related hydrodynamic phonon phenomena in recent years. A prominent example is the Green’s-function solution of the linearized BTE developed by Chiloyan *et al.*¹¹, which explicitly demonstrates that only the full scattering matrix—not the RTA—can correctly describe transient thermal gratings and second sound oscillations in graphene and graphite. This framework has also been employed, directly or implicitly, in the interpretation of experimental TTG observations

of second sound in graphite and other low-dimensional materials. Our theoretical treatment follows this established methodology and is therefore consistent with the current consensus in the field.

Finally, we emphasize that our use of the full scattering matrix BTE is not intended to introduce additional conceptual ambiguity, but rather to avoid it. While the Cattaneo-Vernotte (CV) equation can mathematically model a finite speed of heat propagation, yielding wave-like solutions, its applicability as a physically accurate description of Second Sound in crystalline solids is subject to significant theoretical debate and scrutiny. The traditional derivation of the CV equation critically relies on the assumption that the spatial gradient of the non-equilibrium phonon distribution (δf) is small, which is necessary for applying local-equilibrium expansions that are valid only in the vicinity of the Fourier diffusive limit, i.e., when heat transport is predominantly diffusive and spatial variations occur over length scales much larger than phonon mean free paths. However, the conditions under which Second Sound is typically observed—at micron or submicron length scales (i.e., when $\tau_N \ll \tau_R$)—often lead to large gradients in δf . In such regimes, the very foundation of the CV derivation, which assumes small perturbations from local equilibrium, breaks down. Therefore, the ability of the CV equation to predict wave-like propagation may be considered a mathematical coincidence rather than a genuine reflection of the underlying microscopic phonon hydrodynamics. The implication of the small- δf assumption in the CV equation, and the validity of CV equation for phonon hydrodynamic transport, including second sound, have been discussed in detail¹² (see pp. 258 – 260 and 268 – 273) by Chen. For this reason, we believe that the more fundamental and physically consistent approach, based on the full scattering matrix BTE, is essential for correctly interpreting our experimental results.

We clarify that, in our conclusion, the phrase “super-diffusive enhancement” is not intended to imply fractional or anomalous diffusion in the strict mathematical sense. Rather, it refers to the observation that, within the hydrodynamic transport regime, the effective thermal conductivity associated with collective phonon drift is higher than that in the purely diffusive regime. In this generalized sense, the transport appears faster than Fourier diffusion, but its physical origin remains the emergence of second sound. Importantly, the causal relationship is the opposite of what may have been mistakenly inferred: the enhanced effective thermal conductivity is a consequence of second sound arising from phonon hydrodynamics, not the cause of second sound itself. Microscopically, this enhancement originates from momentum-conserving normal scattering enabling collective phonon motion, while macroscopically it manifests as reduced damping and faster heat propagation in TTG measurements. This physics is naturally captured within the phonon Boltzmann transport equation solved with the full scattering matrix, which is the established and widely accepted theoretical framework for modeling second sound and phonon hydrodynamic transport. For this reason, we do

not adopt fractional transport models, which describe fundamentally different transport mechanisms. We have revised the manuscript to clarify this distinction and to strengthen the discussion of the validity and applicability of the full scattering matrix BTE in the main text and Supplementary Information.

Revisions to the manuscript:

On Results and discussion (Page 9, Lines 219 - 222):

“The full scattering matrix PBTE has been established as a validated and widely accepted theoretical framework for simulating phonon hydrodynamic transport, and a detailed description of this methodology is provided in the section of Details of Numerical Methods.”

On Methods — Details of Numerical Methods (Pages 20 - 21, Lines 634 - 669):

“Second sound is intrinsically a kinetic manifestation of collective phonon dynamics and therefore requires a microscopic transport description beyond Fourier-type diffusion models. Fractional heat transport models and Cattaneo–Vernotte (CV) equation have been proposed to describe anomalous diffusion in systems where energy carriers exhibit Lévy-type flight statistics or long-range spatiotemporal correlations.

In such models, nonlocality in space and/or time is introduced phenomenologically through fractional derivatives, leading to non-Gaussian heat spreading and scale-free superdiffusive behavior. In comparison, the Cattaneo–Vernotte (CV) equation extends Fourier’s law by introducing a finite relaxation time to correct the assumption of instantaneous heat-flux response, resulting in a hyperbolic heat equation with finite propagation speed. Despite these differences, both approaches remain phenomenological and lack an explicit description of microscopic phonon scattering and collective momentum transport, and therefore are not suitable for modeling second sound.

In this work, we adopt the phonon Boltzmann transport equation (BTE), which provides a mode-resolved and time-dependent description of phonon populations and naturally unifies diffusive and wave-like heat transport within a single theoretical framework. It has been well established that second sound emerges directly from the BTE when momentum-conserving normal phonon–phonon scattering dominates over resistive processes, without invoking additional phenomenological assumptions. For this reason, the BTE has long been regarded as the minimal and physically grounded framework for analyzing phonon hydrodynamics and wave-like thermal transport.

Crucially, an accurate description of second sound requires solving the BTE with the full phonon scattering matrix rather than within the relaxation-time approximation (RTA). While computationally convenient, the RTA fails in the hydrodynamic regime by violating conservation laws and suppressing inter-mode coupling associated with

dominant normal scattering, which is the microscopic origin of collective phonon drift and second sound. In contrast, the full scattering matrix formulation rigorously preserves energy and momentum conservation and retains the off-diagonal couplings between phonon modes, enabling a physically correct description of phonon momentum redistribution, collective propagation, and wave damping. Recent theoretical developments, including Green's-function solutions of the linearized BTE, have demonstrated that the full scattering matrix BTE can correctly capture transient thermal gratings and second sound oscillations in materials such as graphite and graphene. As a result, this approach has become the standard and widely accepted theoretical framework for modeling phonon hydrodynamics and second sound in low-dimensional and high-Debye-temperature materials, and it forms the basis of the present theoretical analysis.”

The only real contribution of the manuscript—potentially significant—lies in explaining why the authors detect a wave-like effect at room temperature in graphite, whereas previous TTG studies on graphene (Huberman 2019, Ding 2022) did not. To achieve this, the authors would need to clearly discuss differences in the material itself, isotopic purity, sample thickness, accessible grating-period range, and temporal resolution of the measurements. It is also essential that they provide a quantitative comparison of the obtained heat-propagation speed and thermal relaxation time or damping with previous works, to clarify what is truly new in their experiment and what enables the observation of second sound at room temperature.

Response: We appreciate the referee's comment highlighting the importance of clarifying why a wave-like heat-transport regime is observed at room temperature in our graphite samples, whereas previous TTG studies on graphene did not report such an effect. We fully agree that this point represents a central contribution of our work, and we emphasize that it is rooted in a quantitatively distinct scattering landscape enabled by isotope enrichment, rather than in a purely experimental artifact.

First, we note that the graphite samples investigated in this work exhibit a substantially higher isotopic purity than those used in previous TTG studies. Specifically, the abundance of the ^{12}C isotope is increased from the natural value of approximately 98.9% to 99.93%. This seemingly modest change actually leads to a dramatic suppression of isotope-induced resistive (R) scattering, which is known to play a crucial role in limiting hydrodynamic phonon transport at elevated temperatures. In contrast, earlier TTG experiments on graphite (e.g., Huberman et al.¹³ and Ding et al.¹⁴) were performed on samples with natural isotopic composition, where isotope scattering remains a significant momentum-destroying channel at room temperature.

To quantitatively substantiate this point, we provide a detailed first-principles analysis in Extended Data Fig. 5. As shown in Extended Data Fig. 5a (see below), we present a

statistical distribution of resistive-scattering relaxation times for individual phonon modes, revealing that while most modes undergo R scattering on sub-nanosecond timescales, a non-negligible subset exhibits much longer relaxation times. These long-lived modes are precisely those capable of sustaining phonon momentum over extended periods and thus acting as the primary carriers of hydrodynamic heat flow. Importantly, Extended Data Fig. 5b (see below) demonstrates that isotope scattering rates in isotope-enriched graphite are suppressed by more than an order of magnitude compared to naturally abundant samples. This strong reduction of isotope-induced R scattering shifts the balance decisively in favor of momentum-conserving normal (N) processes.

The combined effect of frequent N scattering and strongly weakened R scattering enables the formation of a quasi-equilibrium phonon distribution described by a displaced Bose–Einstein function^{10,15}, which is the defining microscopic signature of phonon hydrodynamics and second sound. As a result, the temporal and spatial windows over which phonon momentum is approximately conserved are significantly broadened, allowing phonon drift to persist over micrometer length scales and tens of nanoseconds even at room temperature. This regime is inaccessible in samples with natural isotopic abundance due to the undesired strong isotope scattering, providing a clear explanation for why second sound was not observed in earlier TTG measurements on natural graphite under comparable external conditions.

We therefore emphasize that the novelty of our work lies not only in the experimental detection of a wave-like thermal response, but also in the identification—supported by quantitative, mode-resolved first-principles calculations—of isotopic purification as a decisive factor enabling room-temperature second sound in graphite. This microscopic insight directly links material quality to macroscopic transport behavior and clarifies the essential difference between our current experiment and previous TTG studies.

In a word, the isotope purity is the most important factor among those the reviewer suggests us to discuss about. And in our revised manuscript, we've clearly emphasized the high isotope purity value of our synthesized graphite, added more discussion on the influence of phonon-isotope scattering on our observation of second sound, and also provided clear information of the sample thickness, accessible grating-period range, and temporal resolution of the measurements as the reviewer's suggestion.

Revisions to the manuscript:

On Extended Data Fig. 5:

Extended Data Fig. 5. Results of first-principles calculations by ShengBTE. **a**, The distribution of resistive (R) scattering times and normal (N) scattering times for different phonon modes, with the relaxation times mainly ranging from 0 to 0.2 ns at 300 K. **b**, Comparison of mean free path for isotope-enriched graphite (squares) and HOPG (dots) at 125 K, isotopic (I) scattering and normal (N) scattering.

On Results and discussion (Page 6, Lines 158 - 160):

“Millimeter scale small thin flakes with thickness of approximately $150\ \mu\text{m}$, were synthesized. The resulting synthesized graphite possesses a high isotopic purity of 99.93% ^{12}C , providing an ideal platform to minimize extrinsic resistive scattering.”

On Results and discussion (Pages 6 - 7, Lines 191 - 194):

“Our TTG setup provides ~ 100 fs temporal resolution, which is sufficient to capture the ultrafast dynamics, and a tunable grating period (range from $0.95\ \mu\text{m}$ to $54\ \mu\text{m}$), covering the transition from ballistic to hydrodynamic and diffusive regimes.”

On Results and discussion (Page 9, Lines 227 - 231):

“In a word, room temperature drifting second sound is observed in isotope-enriched graphite with TTG measurement, and the results can be well supported by first-principles calculation, demonstrating that the technologically-feasible isotope engineering in graphite (and potentially other materials) indeed opens a new avenue to explore the hydrodynamic wave-like heat transport at room-temperature.”

On Results and discussion (Page 10, Lines 253 - 260):

“Specifically, the presence of ^{13}C isotopes acts as point defects that induce resistive phonon-isotope scattering. In natural graphite (1.10% ^{13}C), this scattering is strong enough to relax phonon momentum, and thereby preventing the formation of second sound at room temperature. By reducing the ^{13}C concentration to 0.07%, this scattering channel is effectively suppressed. The calculated phonon-isotope scattering rate in HOPG is found to be about one order-of-magnitude larger than that of isotope-enriched graphite (Fig. 3a). This significant difference explains why phonon hydrodynamic thermal waves are much less likely to be observed in HOPG at room temperature.”

In its current form, the manuscript combines a solid experimental result with a theoretical narrative that is overly ambitious, imprecise, often conceptually incorrect, and misguided—substantially reducing its scientific value and focus. I suggest that authors consider withdrawing the current manuscript and preparing two clearly focused papers:

- One dedicated to the detection of second sound with a detailed comparison to previous studies.
- Another dedicated to the analysis of the TTG method for determining thermal conductivity, without the introduction of inaccurate microscopic models and the problematic use of the “superdiffusivity” concept. For deeper insight into phonon processes, use physically consistent theoretical approaches to phonon dynamics in 2D materials and their connection to macroscopically measurable parameters. For superdiffusion, use fractional heat-transport models, not conceptually incorrect categorical distinctions.

I also suggest the authors consult the following relevant references:

DOI: [10.1103/RevModPhys.61.41](https://doi.org/10.1103/RevModPhys.61.41)

DOI: [10.1103/PhysRevB.87.115421](https://doi.org/10.1103/PhysRevB.87.115421)

[https://doi.org/10.1016/S1369-7021\(12\)70117-7](https://doi.org/10.1016/S1369-7021(12)70117-7)

<http://dx.doi.org/10.1016/j.spmi.2015.09.027>

<https://doi.org/10.1038/s41467-019-11572-4>

DOI: [10.1103/PhysRevB.107.054311](https://doi.org/10.1103/PhysRevB.107.054311)

DOI: [10.1088/0305-4470/30/21/006](https://doi.org/10.1088/0305-4470/30/21/006)

Mainardi, F. *Fractional Calculus and Waves in Linear Viscoelasticity*, Imperial College Press, 2010

<https://doi.org/10.3390/fractalfract9100653>

<https://doi.org/10.3389/fphy.2017.00052>

Response: We appreciate the referee’s careful and candid assessment of the balance between experiment and theory in the present manuscript. We fully agree that the experimental observation of a wave-like thermal response constitutes the core result of this work, and we regret if the breadth of the theoretical discussion may have obscured this central message in its current form, or may have caused any misunderstanding. We wish to clarify that our intention was not to advance a new or speculative theoretical framework, but rather to interpret the experimental findings within a well-established microscopic description of phonon transport.

In this context, we would like to clarify that the theoretical analysis is deliberately based on the phonon Boltzmann transport equation with the full scattering matrix, which represents the current state-of-the-art and widely accepted framework for describing second sound and hydrodynamic phonon transport, as we discussed above. Rather than being conceptually misguided, this approach was chosen precisely to avoid

phenomenological assumptions and to maintain a direct connection between microscopic scattering processes and macroscopic observables (i.e., the dynamics of collective phonon energy) measured in TTG experiments.

Following the referee's insightful comments, we have streamlined the theoretical narrative to focus on its essential role: explaining why the experimental conditions realized here—most notably the strongly suppressed resistive scattering enabled by isotope enrichment—allow second sound to persist at room temperature, in contrast to previous TTG studies. As we discussed and noted earlier, we have made careful revision according to the reviewer's comments and cited relevant references (Ref. 20, 23 – 24 and 37 – 45) suggested by the reviewer in Methods. We believe the refined theoretical and experimental components in our revised manuscript now mutually reinforce and corroborate each other, seamlessly weaving together into a coherent and compelling narrative. With these clarifications and revisions, the theoretical analysis in the revised manuscript now serves as a coherent and physically grounded complement to the experimental results, thereby enhancing rather than detracting from the scientific value and the focus of the manuscript.

Revisions to the manuscript:

On Results and discussion (Page 14, Lines 362 - 365) [Ref. 20, 23, 24]:

“In the literatures, extensive studies have been devoted to the thermal conductivity of graphite and graphene [Ref. 20, 23, 24]; however, the impact of hydrodynamic phonons on thermal conductivity enhancement has rarely been revealed and discussed,”

On the References:

[Ref. 20] Alofi, A. & Srivastava, G. P. Thermal conductivity of graphene and graphite. *Phys. Rev. B* **87**, 115421 (2013).

[Ref. 23] Jaćimovski, S. K., Bukurov, M., Štrajčić, J. P. & Raković, D. I. Phonon thermal conductivity of graphene. *Superlattices Microstruct.* **88**, 330–337 (2015)

[Ref. 24] Balandin, A. A. & Nika, D. L. Phononics in low-dimensional materials. *Mater. Today* **15**, 266–275 (2012).

On Methods — Details of Numerical Methods (Page 20, Lines 634 - 638) [Ref. 37 - 39]:

“Second sound is intrinsically a kinetic manifestation of collective phonon dynamics and therefore requires a microscopic transport description beyond Fourier-type diffusion models. Fractional heat transport models and Cattaneo–Vernotte (CV) equation have been proposed to describe anomalous diffusion in systems where energy carriers exhibit Lévy-type flight statistics or long-range spatiotemporal correlations [Ref. 37 - 39].”

On the References:

[Ref. 37] Compte, A. & Metzler, R. The generalized Cattaneo equation for the description of anomalous transport processes. *J. Phys. Math. Gen.* **30**, 7277 (1997).

[Ref. 38] ateishi, A. A., Ribeiro, H. V. & Lenzi, E. K. The Role of Fractional Time-Derivative Operators on Anomalous Diffusion. *Front. Phys.* **5**, (2017).

[Ref. 39] Galovic, S. *et al.* Electrical Analogy Approach to Fractional Heat Conduction Models. *Fractal Fract.* **9**, (2025).

On Methods — Details of Numerical Methods (Page 20, Lines 639 - 646) [Ref. 40 - 42]:

“In such models, nonlocality in space and/or time is introduced phenomenologically through fractional derivatives, leading to non-Gaussian heat spreading and scale-free superdiffusive behavior [Ref. 40]. While these approaches are suitable for capturing stochastic transport dominated by rare long free paths [Ref. 41 - 42], they do not explicitly account for the microscopic phonon scattering mechanisms or the collective momentum-conserving dynamics that underlie second sound.”

On the References:

[Ref. 40] Mainardi, F. *Fractional Calculus and Waves in Linear Viscoelasticity: An Introduction to Mathematical Models.* (IMPERIAL COLLEGE PRESS, 2010).

[Ref. 41] Isaeva, L., Barbalinardo, G., Donadio, D. & Baroni, S. Modeling heat transport in crystals and glasses from a unified lattice-dynamical approach. *Nat. Commun.* **10**, 3853 (2019).

[Ref. 42] Fiorentino, A. & Baroni, S. From Green-Kubo to the full Boltzmann kinetic approach to heat transport in crystals and glasses. *Phys. Rev. B* **107**, 054311 (2023).

On Methods — Details of Numerical Methods (Pages 20 - 21, Lines 647 - 654) [Ref. 43, 44]:

“In this work, we adopt the phonon Boltzmann transport equation (BTE), which provides a mode-resolved and time-dependent description of phonon populations and naturally unifies diffusive and wave-like heat transport within a single theoretical framework. It has been well established that second sound emerges directly from the BTE when momentum-conserving normal phonon–phonon scattering dominates over resistive processes, without invoking additional phenomenological assumptions. For this reason, the BTE has long been regarded as the minimal and physically grounded framework for analyzing phonon hydrodynamics and wave-like thermal transport [Ref. 43, 43].”

On the References:

[Ref. 43] Ward, A., Broido, D. A., Stewart, D. A. & Deinzer, G. Ab initio theory of the lattice thermal conductivity in diamond. *Phys. Rev. B* **80**, 125203 (2009).

[Ref. 44] Lindsay, L., Broido, D. A. & Mingo, N. Flexural phonons and thermal transport in graphene. *Phys. Rev. B* **82**, 115427 (2010).

On Methods — Details of Numerical Methods (Page 21, Lines 655 - 669):

“Crucially, an accurate description of second sound requires solving the BTE with the full phonon scattering matrix rather than within the relaxation-time approximation (RTA). While computationally convenient, the RTA fails in the hydrodynamic regime by violating conservation laws and suppressing inter-mode coupling associated with dominant normal scattering, which is the microscopic origin of collective phonon drift and second sound. In contrast, the full scattering matrix formulation rigorously preserves energy and momentum conservation and retains the off-diagonal couplings between phonon modes, enabling a physically correct description of phonon momentum redistribution, collective propagation, and wave damping. Recent theoretical developments, including Green’s-function solutions of the linearized BTE, have demonstrated that the full scattering matrix BTE can correctly capture transient thermal gratings and second sound oscillations in materials such as graphite and graphene. As a result, this approach has become the standard and widely accepted theoretical framework for modeling phonon hydrodynamics and second sound in low-dimensional and high-Debye-temperature materials, and it forms the basis of the present theoretical analysis. We now briefly describe the formalism of the full scattering matrix BTE we employed.”

On Methods — Analysis of TTG Signal to obtain second sound speed, propagation length, and the effective k (Page 25, Line 773 - 774) [Ref. 45]:

“According to the classical Fourier Law of heat conduction, the heat transfer process is driven by the temperature gradient [Ref. 45].”

On the References:

[Ref. 45] Joseph, D. D. & Preziosi, L. Heat waves. *Rev. Mod. Phys.* **61**, 41–73 (1989).

Reviewer #3:

This is an impressive experimental achievement. The authors observed second sound at room temperature in graphite, a result considered nearly impossible due to the extreme sensitivity of second sound to the isotopic purity of the material. The success was made possible by the preparation of a highly isotopically pure sample.

This work is fundamentally important because it demonstrates that phonon hydrodynamics can be accessed and exploited at room temperature. The observation of second sound in this context is a significant step forward, as it opens the door to practical applications involving materials with high thermal conductivity. Given these advancements, this work is certainly deserving of publication.

Response: We thank you very much for your thorough review and for recognizing the importance of our work, as well as for your recommendation for publication.

References in Response Letter

1. Stern, M. J. *et al.* Mapping momentum-dependent electron-phonon coupling and nonequilibrium phonon dynamics with ultrafast electron diffuse scattering. *Phys. Rev. B* **97**, 165416 (2018).
2. Kremeyer, L., Britt, T. L., Siwick, B. J. & Huberman, S. C. Ultrafast electron diffuse scattering as a tool for studying phonon transport: Phonon hydrodynamics and second sound oscillations. *Struct. Dyn.* **11**, 024101 (2024).
3. Beardo, A. *et al.* Observation of second sound in a rapidly varying temperature field in Ge. *Sci. Adv.* **7**, eabg4677 (2021).
4. Yue, S. *et al.* High ambipolar mobility in cubic boron arsenide revealed by transient reflectivity microscopy. *Science* **377**, 433–436 (2022).
5. Lee, S., Broido, D., Esfarjani, K. & Chen, G. Hydrodynamic phonon transport in suspended graphene. *Nat. Commun.* **6**, 6290 (2015).
6. Simoncelli, M., Marzari, N. & Cepellotti, A. Generalization of Fourier’s Law into Viscous Heat Equations. *Phys. Rev. X* **10**, 011019 (2020).
7. Johnson, J. A. *et al.* Direct Measurement of Room-Temperature Nondiffusive Thermal Transport Over Micron Distances in a Silicon Membrane. *Phys. Rev. Lett.* **110**, 025901 (2013).
8. Ward, A., Broido, D. A., Stewart, D. A. & Deinzer, G. Ab initio theory of the lattice thermal conductivity in diamond. *Phys. Rev. B* **80**, 125203 (2009).
9. Lindsay, L., Broido, D. A. & Mingo, N. Flexural phonons and thermal transport in graphene. *Phys. Rev. B* **82**, 115427 (2010).

-
10. Hardy, R. J. Phonon Boltzmann Equation and Second Sound in Solids. *Phys. Rev. B* **2**, 1193–1207 (1970).
 11. Chiloyan, V. *et al.* Green's functions of the Boltzmann transport equation with the full scattering matrix for phonon nanoscale transport beyond the relaxation-time approximation. *Phys. Rev. B* **104**, 245424 (2021).
 12. Chen, G. & Chen, G. *Nanoscale Energy Transport and Conversion: A Parallel Treatment of Electrons, Molecules, Phonons, and Photons*. (Oxford University Press, Oxford, New York, 2005).
 13. Huberman, S. *et al.* Observation of second sound in graphite at temperatures above 100 K. *Science* **364**, 375–379 (2019).
 14. Ding, Z. *et al.* Observation of second sound in graphite over 200 K. *Nat. Commun.* **13**, 285 (2022).
 15. Cepellotti, A. *et al.* Phonon hydrodynamics in two-dimensional materials. *Nat. Commun.* **6**, 6400 (2015).

Authors' Response to Reviewer Comments

We sincerely thank the Reviewers for their careful and constructive evaluations of our manuscript. In response to their suggestions, we have further expanded the discussion of the experimental results, refined the theoretical framework, and comprehensively revised the manuscript. We believe these revisions have substantially strengthened the quality and impact of the study. Below, we provide detailed, point-by-point responses to all reviewers' comments (reviewer comments are presented in **blue**, our response is presented in **black**).

Reviewer #1:

The Authors provided detailed responses to all my comments. Their responses were accompanied by appropriate revisions of the manuscript and inclusion of additional extended data that I found very informative. I also appreciate their detailed explanations regarding some general questions I had.

At this stage, I do not have additional comments or questions.

The observation of second sound at room temperature - which is the main experimental finding of this work - is interesting and deserves publication. The Authors have provided enough data and a detailed discussion of their experimental and theoretical methods so that others can reproduce or extend their work. I believe that this article will attract attention from the community and trigger further developments. Therefore, I recommend publication.

Response: We sincerely thank the reviewer for the highly positive and encouraging assessment of our work. We greatly appreciate the careful evaluation of the revised manuscript and the recognition of both the experimental significance and the clarity of the theoretical and methodological discussions. We are pleased that the additional data and detailed explanations were found informative and that the reproducibility and potential impact of our results are acknowledged. We believe that the reviewer's comments have helped improve the quality and presentation of the manuscript, and we are grateful for their support and recommendation for publication.

Reviewer #2:

The manuscript presents a very strong experimental contribution with original results that are significant for the field. The revised version substantially improves the explanation of the experimental setup. It is now clearly shown that the authors, thanks to the improved temporal and spatial resolution of the TTG setup and the use of isotopically purified graphite, were able to detect the oscillatory signal interpreted as second sound at room temperature. This part of the work is very strong and scientifically valuable.

The authors have also addressed the first-round comment by removing the term super-diffusive, avoiding serious terminological confusion that could have misled readers regarding the model used.

Response: We sincerely thank the reviewer for these positive and encouraging comments. We are pleased that the revised manuscript has clarified the experimental methodology and that the significance of the improved temporal and spatial resolution, together with isotopic purification, in enabling the observation of oscillatory second-sound-like signals at room temperature is now clearly recognized. We also appreciate the reviewer's acknowledgement that the removal of the term "super-diffusive" has improved the clarity and avoided potential terminological confusion. We believe these revisions have strengthened the focus and scientific impact of the manuscript.

However, despite these significant improvements, there remains a fundamental issue in the manuscript's structure and interpretation that requires clarification.

1. Role of ab initio and BTE in relation to the experimental analysis

From the manuscript structure, it appears that the ab initio / BTE results are not directly used. The workflow can be summarized as follows:

- ab initio and linearized BTE are used to obtain the temperature response in the Fourier domain (Eq. 1–5);
- through further simplifications and the Callaway approximation, a system is obtained that reduces to a modified telegraph (Cattaneo–Vernotte type) equation (Eq. 33);
- Eq. (33) is then used to interpret the TTG signal and fit the oscillatory behavior;
- The effective conductivity k_{eff} is determined by additional empirical fitting of the exponential decay of the signal via $k=c\Gamma/q^2$.

From this procedure, it is clear that in the analysis of the experimental data (Figs. 1–4), the results of ab initio / BTE calculations are not directly used; rather, a reduced macroscopic model and parameters obtained from fitting the TTG signal are employed. However, the manuscript does not make this distinction sufficiently clear, leaving the quantitative role of the ab initio / BTE part in the experimental interpretation ambiguous.

Response: We thank the reviewer for this careful reading and for pointing out that the role of the ab initio and BTE calculations in relation to the experimental analysis was not sufficiently clearly articulated in the manuscript. We agree that the current

presentation may give the misleading impression that the TTG signals are interpreted or fitted using the reduced Callaway-based macroscopic model. This, however, is not the case, and we clarify the workflow below.

First, we emphasize that the simulated TTG signals and results shown in Figs. 1 – 4 are not obtained from Eq. (33) or from the Callaway approximation. Actually, all theoretical TTG signals are calculated directly from the full scattering matrix phonon Boltzmann transport equation, as formulated in Eq. (5). These calculations explicitly combine ab initio density functional theory (DFT) inputs—including phonon dispersions, group velocities, mode-dependent heat capacities, and anharmonic interaction strengths—with the temperature- and isotope-dependent full phonon scattering matrix obtained from the BTE. This DFT+BTE framework is essential for quantitatively simulating the spatiotemporal evolution of the transient thermal grating under isotopic purification conditions. Importantly, every measured TTG figure in the manuscript contains a corresponding theoretical curve computed in this way, and the excellent quantitative agreement with experiment is achieved without any fitting parameters. These quantitative results are not reproduced by the Callaway model and are exclusively obtained from the full scattering matrix BTE.

In contrast, the Callaway model is introduced solely for the qualitative interpretation of the effective thermal conductivity extracted from the TTG signals. As correctly noted by the reviewer, the effective conductivity k_{eff} is determined by fitting the exponential decay of the TTG signal via $k=c\Gamma/q^2$. We apply this fitting procedure to the theoretically calculated TTG signals and find that, in the hydrodynamic transport regime, an apparent enhancement of k_{eff} emerges. To provide physical insight into this observation, we use the Callaway framework to qualitatively illustrate how the coexistence of dominant normal scattering and suppressed resistive scattering leads to an enhanced hydrodynamic contribution to thermal transport. Specifically, we performed a derivation starting from the Callaway-modified BTE (double relaxation-time-approximated BTE), and showed that the effective thermal conductivity is actually composed of two parts: $k = k_K + k_H$, where k_K is the kinetic part corresponding to the phonon diffusion, while k_H is the hydrodynamic part corresponding to phonon hydrodynamics. And this k_H term qualitatively explains the hydrodynamic enhancement in the effective thermal conductivity. In short, the Callaway model does not enter any part of the quantitative TTG simulations, nor does it determine the oscillation frequency, damping, or phase of the TTG response.

We would like to kindly point out that our previous manuscript, to a certain extent, has covered the relationship between ab initio BTE and experimental results in several places. For instance, after the introduction of the full scattering matrix PBTE, we compared the experiment and simulated results, and mentioned “Excellent agreement between the experiments and the simulation is achieved, with the reappearance of the

hydrodynamic sign flipping feature for the isotope-enriched graphite and the diffusive decay feature for HOPG accordingly”. And “This suggests that both the speed and the strength of second sound, can be mutually confirmed and acquired by the experiment and the PBTE theory”. In the section discussing the temperature and the length dependences of TTG signals, we also noted “Each sub-figure of Fig. 2 displays the experimental and the simulated TTG signals under three temperatures. The experiments and the simulations agree well, showing the same trending pattern during the temperature or grating period changes”.

To enhance the clarity regarding the roles of the two theoretical models, ab initio PBTE and Callaway BTE, we have further provided some additional explanation and refined the relevant descriptions, making sure we have clearly separated their two roles: (i) the full scattering matrix DFT+BTE framework as the sole quantitative tool used to compute TTG signals and compare directly with experiment, and (ii) the Callaway model as a complementary, qualitative framework used only to aid physical interpretation of the extracted effective thermal conductivity. We believe this clarification resolves the ambiguity identified by the reviewer and strengthens the logical structure of the manuscript.

Revisions to the manuscript:

On Main text (Pages 17, Lines 375 - 380):

“This approach provides a clear and intuitive physical picture. Starting from the Callaway-modified phonon Boltzmann transport equation, we first derive two coupled viscous heat equations, analogous to those obtained from the linearized BTE within the relaxon framework. These viscous heat equations can then be further reduced to a hyperbolic form (or namely thermal telegraph equation)[Ref. 6, 26, 27], which is formally identical to the Cattaneo–Vernotte (CV) equation)[Ref. 28, 29].”

On the References (new reference only):

[Ref. 25] J. Callaway, Model for lattice thermal conductivity at low temperatures, *Phys. Rev.* **113** (4) (1959) 1046–1051.

[Ref. 26] Qian, X. *et al.* API phonons: Python interfaces for phonon transport modeling. *Mater. Today Phys.* **50**, 101630 (2025).

[Ref. 27] Qian, X. *et al.* Analytical Green's function of the multidimensional Boltzmann transport equation for modeling hydrodynamic second sound. *Phys. Rev. B* **111**, 035406 (2025).

On Methods — Details of Numerical Methods (Pages 27, Lines 712 - 715):

“Equation (5) represents the frequency-domain response of the temperature field. Consequently, the temporal dynamics of the TTG signal can be obtained by performing an inverse Fourier transform of Eq. (5), which corresponds to the product of the heat source and the Green’s function evaluated at the wave vector q associated with the TTG

period.”

2. Interpretation of the results in Figures 3 and 4

The results in Figures 3 and 4 are very interesting, but their interpretation through the decomposition $k_{\text{eff}}=k_K+k_H$ and the division into “hydrodynamic”, “ballistic”, and “diffusive” contributions gives the impression that the experiment directly separates these contributions. According to the methodology, k_{eff} is derived from the time decay of the TTG signal, while its decomposition relies on an additional theoretical construction not directly linked to the fitting procedure or to the ab initio / BTE part.

Response: We appreciate the reviewer’s insightful comment and agree that the experiment itself does not directly separate the hydrodynamic, ballistic, and diffusive contributions to heat transport. The effective thermal conductivity k_{eff} is obtained solely from the exponential decay of the TTG signal, and the experimental analysis does not provide a direct decomposition of k_{eff} into different transport channels; however, by analyzing the size dependence of k_{eff} as a function of the grating period, we can naturally identify three distinct transport regimes on the basis of their characteristic transport length scales.

We clarify again that the decomposition $k_{\text{eff}} = k_K + k_H$ is derived result from the Callaway model as a qualitative interpretative framework , rather than quantities or relation directly extracted from the experiment or from the fitting procedure. This theoretical decomposed thermal conductivity expression is therefore not directly linked to the ab initio or full scattering matrix BTE calculations, but is instead from Callaway model and used to provide physical intuition for the observed trends in k_{eff} , particularly the apparent enhancement in the hydrodynamic regime.

In other words, while the full scattering matrix DFT+BTE framework is employed to quantitatively compute the TTG signals and reproduce the experimental observations, the Callaway model is invoked only at a conceptual level to rationalize how the competition between normal and resistive scattering can lead to an enhanced effective thermal conductivity. We have revised the manuscript to explicitly state that this decomposition is not an experimental separation, but a qualitative theoretical interpretation, in order to avoid any potential misunderstanding.

Regarding to the identification of three different phonon transport regimes, this is done by the inspection on the transport-length dependence of the effective thermal conductivity, namely, the thermal conductivity size effect. It is well known that, as the transport length L decreases from larger than phonon MFP (i.e., $L>\text{MFP}$) to smaller than phonon MPF (i.e., $L<\text{MFP}$), the effective thermal conductivity would transition from the diffusive plateau and gradually decrease toward zero, following the ballistic behavior: $k_{\text{ballistic}} \sim L$. And if a material system can support hydrodynamic phonon

transport, the hydrodynamic behavior happens to take place in the diffusive-ballistic transition region, because hydrodynamic transport requires $MFP_{\text{normal}} < L < MPF_{\text{resistive}}$. This is why one can naturally identify three distinct transport regimes according to the characteristics of the extracted effective thermal conductivity on transport length scales.

Revisions to the manuscript:

On Results and discussion (Page 17, Lines 375 - 380):

“This approach provides a clear and intuitive physical picture. Starting from the Callaway-modified phonon Boltzmann transport equation, we first derive two coupled viscous heat equations, analogous to those obtained from the linearized BTE within the relaxon framework. These viscous heat equations can then be further reduced to a hyperbolic form (or namely thermal telegraph equation)[Ref. 6, 26, 27], which is formally identical to the Cattaneo–Vernotte (CV) equation[Ref. 28, 29].”

3. Terminology issues

The use of terms such as hydrodynamic phonons, ballistic phonons, diffusive phonons, and collective phonons may lead to confusion. In standard terminology, these terms refer to transport regimes rather than phonon types. In the manuscript, they are used in a way that suggests a different classification, which requires explicit clarification—especially given their impact on the interpretation of the results in Figures 3 and 4.

Response: We thank the reviewer for raising this important terminology point and agree that terms such as “hydrodynamic,” “ballistic,” and “diffusive” should strictly refer to transport regimes rather than to distinct phonon species. We clarify that in our manuscript these terms are not intended to define any specific phonon mode which is theoretically identified by its frequency ω and wavevector q , as possessing intrinsic, permanent, time-invariant diffusive or hydrodynamic or ballistic characters, but rather to describe the dominant transport behavior of the phonon population under different scattering conditions and length–time scales. Strictly speaking, phonon modes are defined by their ω - q dispersion relations. However, phonons belonging to different modes may exhibit hydrodynamic, ballistic, or diffusive transport behavior depending on temperature and the grating period (i.e., the relevant transport length scale). As a result, in a given transport regime, phonons would predominantly or typically exhibit the corresponding transport characteristics, although a small fraction of phonons may simultaneously display behaviors associated with other regimes. This regime-based description has been widely adopted in the literatures¹⁻³ to convey the underlying physical picture in a clear and intuitive manner, and we therefore consider it appropriate to retain this terminology in the present work, while explicitly clarifying its intended meaning.

Specifically, by “hydrodynamic phonons” or “collective phonons” we refer to phonons whose transport is governed by strong momentum-conserving normal scattering,

leading to collective drift and wave-like heat propagation, as in the second-sound regime. Likewise, the term “ballistic phonons” denotes phonons whose mean free paths exceed the relevant length scales of the experiment, resulting in weakly scattered, quasi-ballistic transport, while “diffusive phonons” refers to phonons undergoing frequent resistive scattering and therefore contributing predominantly to Fourier-like diffusion. These descriptors characterize transport regimes or dynamical behavior, not intrinsic phonon identities.

Key concern

The way the theoretical part is connected to the experimental analysis could lead readers to incorrect conclusions about what is actually measured and what represents the theoretical framework for interpretation. This appears primarily as a structural and terminological issue rather than a problem with the results themselves.

Recommendation for clarification (without changing the results)

1. Clearly separate the experimental analysis based on Eq. (33) from the theoretical context based on ab initio / BTE calculations.
2. Precisely explain how k_{eff} is obtained and interpreted.
3. Provide clearer definitions for the terminology used regarding transport regimes and phonons.

Conclusion

The work presents a strong experimental contribution, but a major revision is necessary to clarify the narrative, terminology, and the relation between the theoretical and experimental parts of the manuscript.

Response: We thank the reviewer for clearly articulating the key concern and for recognizing that the issue primarily lies in the structure and terminology of the manuscript rather than in the experimental results themselves. We agree that, in its previous form, the connection between the theoretical framework and the experimental analysis could have led to ambiguity regarding what is directly measured and what serves as theoretical interpretation.

In response to the reviewer’s recommendations, we have revised the manuscript and thoroughly checked it to ensure that the different components of the analysis are now clearly separated. First, we explicitly distinguish the experimental extraction of the TTG decay and oscillatory features from the theoretical modeling based on ab initio and full scattering matrix BTE calculations. The role of Eq. (33) is now clearly stated as a reduced macroscopic model used only for the analysis of effective thermal conductivity, while all quantitative TTG signals are obtained directly from the full scattering matrix BTE.

Second, we have revised the text to precisely explain how the effective thermal conductivity k_{eff} is obtained from the exponential decay of the TTG signal and to clarify

that its interpretation relies on a qualitative theoretical framework rather than a direct experimental separation of transport contributions.

Third, we have carefully revised the terminology throughout the manuscript to ensure that terms such as hydrodynamic, ballistic, and diffusive consistently refer to transport regimes or dominant scattering behaviors, rather than to distinct phonon types. Explicit definitions have been added to avoid any potential misunderstanding.

Importantly, these revisions do not alter any experimental data, theoretical calculations, or conclusions. Instead, they clarify the narrative and strengthen the logical connection between the experimental observations and their theoretical interpretation. We believe that these changes address the reviewer's concerns and significantly improve the clarity and readability of the manuscript.

Reviewer #3:

I believe that the authors addressed all Referee's remarks, so the manuscript is ready for publication.

Response: We sincerely thank the reviewer for this highly positive assessment. We appreciate the careful evaluation of the revised manuscript and are pleased that the responses and revisions were found to adequately address all of the referee's remarks. We are grateful for the reviewer's support and recommendation for publication.

References in Response Letter

1. Ding, Z. *et al.* Observation of second sound in graphite over 200 K. *Nat. Commun.* **13**, 285 (2022).
2. Pumarol, M. E. *et al.* Direct Nanoscale Imaging of Ballistic and Diffusive Thermal Transport in Graphene Nanostructures. *Nano Lett.* **12**, 2906–2911 (2012).
3. Vakulov, D. *et al.* Ballistic Phonons in Ultrathin Nanowires. *Nano Lett.* **20**, 2703–2709 (2020).

Authors' Response to Reviewer Comments

We would like to express our sincere gratitude to the reviewer for his/her insightful and constructive comments. We are particularly encouraged by the reviewer's recognition of our experimental results as a "valuable contribution" and the confirmation that our observations regarding isotope purification in graphite are solid and important.

Regarding the theoretical framework used to interpret our data, the reviewer raises a profound point concerning the limitations of the heuristic hydrodynamic/ballistic/diffusive picture. We offer the following clarifications:

We agree with the reviewer that "effective thermal conductivity" is an interpreted quantity. In this work, the effective thermal conductivity we report is derived by fitting our simulated TTG signal, which is from the solution of the Peierls-Boltzmann Transport Equation (PBTE), with an exponential decay function. This treatment is valid and widely accepted when it comes to extracting the effective decay rate (the effective k). Moreover, while the PBTE itself has limitations within the broader context of non-equilibrium thermodynamics, it remains the standard computational benchmark for phonon transport in crystalline solids.

In addition, we utilize the Callaway-type arguments primarily as a qualitative heuristic to provide physical intuition for the observed trends. We acknowledge that while this model is widespread in the literature, it simplifies the complex wave-like nature of thermal transport into a more intuitive "phonon gas" framework.

We believe that by grounding our quantitative analysis in first-principle PBTE calculation and using the Callaway model for qualitative explanation, our data processing remains both robust and accessible to the current community. We thank the reviewer again for their rigorous oversight, which has helped clarify the theoretical positioning of our work.